# Hazard Assessment of the Effects of Acute and Chronic Exposure to Permethrin, Copper Hydroxide, Acephate, and Validamycin Nanopesticides on the Physiology of *Drosophila*: Novel Insights into the Cellular Internalization and Biological Effects

**DOI:** 10.3390/ijms23169121

**Published:** 2022-08-14

**Authors:** Eşref Demir, Seyithan Kansız, Mehmet Doğan, Önder Topel, Gökhan Akkoyunlu, Muhammed Yusuf Kandur, Fatma Turna Demir

**Affiliations:** 1Medical Laboratory Techniques Program, Vocational School of Health Services, Department of Medical Services and Techniques, Antalya Bilim University, Antalya 07190, Turkey; 2Faculty of Science, Department of Chemistry, Akdeniz University, Antalya 07070, Turkey; 3Faculty of Science, Department of Chemistry, Ankara University, Ankara 07100, Turkey; 4Faculty of Medicine, Department of Histology and Embryology, Akdeniz University, Antalya 07070, Turkey; 5Department of Histology and Embryology, Faculty of Medicine, Kırklareli University, Kırklareli 39100, Turkey; 6Industrial Biotechnology and Systems Biology Research Group, Faculty of Engineering, Department of Bioengineering, Marmara University, İstanbul 34854, Turkey

**Keywords:** *Drosophila melanogaster*, nanopesticides, permethrin nanopesticides, copper hydroxide nanopesticides, acephate nanopesticides, validamycin nanopesticides, internalization, gene expression, oxidative stress, DNA damage, lipid peroxidation, phenotypic variations, locomotor ability, nanogenotoxicity, risk assessment

## Abstract

New insights into the interactions between nanopesticides and edible plants are required in order to elucidate their impacts on human health and agriculture. Nanopesticides include formulations consisting of organic/inorganic nanoparticles. *Drosophila melanogaster* has become a powerful model in genetic research thanks to its genetic similarity to mammals. This project mainly aimed to generate new evidence for the toxic/genotoxic properties of different nanopesticides (a nanoemulsion (permethrin nanopesticides, 20 ± 5 nm), an inorganic nanoparticle as an active ingredient (copper(II) hydroxide [Cu(OH)_2_] nanopesticides, 15 ± 6 nm), a polymer-based nanopesticide (acephate nanopesticides, 55 ± 25 nm), and an inorganic nanoparticle associated with an organic active ingredient (validamycin nanopesticides, 1177 ± 220 nm)) and their microparticulate forms (i.e., permethrin, copper(II) sulfate pentahydrate (CuSO_4_·5H_2_O), acephate, and validamycin) widely used against agricultural pests, while also showing the merits of using *Drosophila*—a non-target in vivo eukaryotic model organism—in nanogenotoxicology studies. Significant biological effects were noted at the highest doses of permethrin (0.06 and 0.1 mM), permethrin nanopesticides (1 and 2.5 mM), CuSO_4_·5H_2_O (1 and 5 mM), acephate and acephate nanopesticides (1 and 5 mM, respectively), and validamycin and validamycin nanopesticides (1 and 2.5 mM, respectively). The results demonstrating the toxic/genotoxic potential of these nanopesticides through their impact on cellular internalization and gene expression represent significant contributions to future nanogenotoxicology studies.

## 1. Introduction

Scientists and researchers are desperately working to come up with new ways to increase food production, as the global population is projected to grow by about 48% by 2050 [1]. Shrinking agricultural areas and increased food demand in parallel with swelling urban populations lead to large-scale monoculture of many crops that cannot be sustained without the use of agrochemicals. The global market size of agrochemicals, including pesticides, fertilizers, and growth hormones, was estimated to reach USD 266 billion by 2021, at an annual growth rate of 4.5% [2]. Despite the benefits of the agricultural food sector, widespread use of agrochemicals also causes large amounts of hazardous chemicals to escape into a range of natural habitats [3]. Additional use of pesticides to control pest (insect) resistance further contaminates terrestrial and aquatic ecosystems, along with toxic residues in foods grown for human or animal consumption [4]. The annual death toll due to ingestion of pesticides through such foods is around 20,000, as they interact with the microbiome in the human gastrointestinal tract, causing digestive problems, lung cancer, and hormonal imbalances [5].

The key concept in nanopesticide formulations involves improving the delivery and efficacy of the active ingredient(s) through nanoscale particles [6]. However, the term “nanopesticide” more commonly refers to controlled release systems, where the active ingredients are often encapsulated by biodegradable nanocarriers [7]. The term is also used to describe formulations consisting of organic or inorganic nanoparticles (NPs) in which the particle itself or its dissolved form has the desired activity [8], or hybrid formulations combining inorganic and organic materials [9]. Nanopesticides often contain nanomaterials (NMs), which are used as carriers of pesticides to ensure an efficiently controlled release at more effective concentrations, and such NMs include a wide variety of products that combine various surfactants, capsules, metal oxides, particles, and polymers at the nano scale (usually 1–100 nm in size) [10].

Research into certain applications of nanotechnology in agriculture has become increasingly popular over the last decade, with a particular interest in nanopesticides [5,6,10,11,12,13,14,15,16,17,18]. Engineering of novel plant protection products has been gaining more attention than other applications, such as nanosensors or fertilizers. Although there exist several reports on a few examples of nanopesticides in the relevant literature [6,7,19,20,21], an accurate assessment of the current status in this field appears to be rather challenging. One reason for this is that there is currently no precise and agreed-upon definition for a nanopesticide. Some articles consider formulations of particles up to 1000 nm to be nanopesticides, which contradicts the typical 1–100 nm size range generally accepted for NPs (ISO/TS 80004-2:2015). Another reason is that even if some articles differ in their mode of action and target species, nanoforms of both pesticides and biocides are referred to as nanopesticides.

The main objective of nanoemulsions is usually to enhance the apparent solubility of poorly soluble active ingredients (AIs) while keeping the concentration of surfactants lower than that of the microemulsion (typically 5–10% surfactant compared to 20% microemulsion). Nanoemulsions can be prepared by using certain active ingredients, such as neem [22], permethrin [23], and glyphosate [24,25,26]. Nanoemulsions have garnered considerable popularity in the pharmaceutical industry as, for instance, potential carriers for the transdermal delivery of hydrophobic drugs, which exhibit poor solubility in water [27]. Although it has been suggested that nanoemulsions with pesticide active ingredients increase the uptake of active ingredients, evidence for their safe use in plant protection products is still insufficient. However, two recent studies support the hypothesis that they can accelerate the uptake of active ingredients. In the first of these studies, experiments with nanoemulsions containing neem oil revealed that the LC_50_ (i.e., the concentration required to achieve 50% mortality) decreased with droplet size, pointing to an increase in the uptake of smaller droplets [22]. In some of the recent studies in the literature, hemocytes were used to monitor living organisms in nature under environmental conditions, using new techniques (such as flow cytometry) [28,29].

Permethrin is a synthetic pyrethroid insecticide with a saturated aqueous solubility of 0.2 mg/L at 25 °C, and is poorly soluble in water [30]. The findings published by Anjali et al. [23] were interpreted as nanoemulsions increasing the uptake of active ingredients. In this second study, their effects on non-target organisms (e.g., soil bacteria and plants) were found to be decreased [23], but the reasons for their different effects on target and non-target organisms have yet to be fully elucidated. Unfortunately, no comparisons with commercial formulations or pure active ingredients have been made either. Our study evaluating the effect of permethrin alone on *Drosophila melanogaster* showed that the lethal concentrations for permethrin were LC_25_ = 22.5 ppm and LC_50_ = 46.47 ppm [31]. In another study, we tested permethrin alone at the LC_25_ dose, and in combination with different ratios of piperonyl butoxide (PBO, the most widely used product among the methylenedioxyphenyl group synergists); however, no genotoxic activity was observed in either application group in the *Drosophila* wing somatic mutation and recombination test (SMART) [32]. Sundaramoorthy et al. [33] investigated the in vitro toxic and genotoxic effects of permethrin at microparticle and nanoparticle size on human peripheral erythrocytes and lymphocytes through erythrocyte morphology tests, cell viability tests, and micronucleus tests. They discovered that both permethrin forms caused toxic effects and chromosome damage in erythrocytes/lymphocytes. Kumar et al. [23], who tested the toxicity of nanopermethrin, observed that nanopermethrin had a greater toxic effect than permethrin at lower concentrations in the target species (*Aedes aegypti*), while in non-target bacteria and plant species (e.g., *Escherichia coli*, *Bacillus subtilis*, *Cucumis sativus*, *Zea mays*, *Lycopersicum esculentum*, and *Allium cepa)* nanopermethrin caused fewer toxic effects than bulk permethrin. Based on their findings, we can conclude that the formulated nanopermethrin affords a higher efficiency and better biosafety compared to permethrin. Similarly, Mishra et al. [34] reported that nanopermethrin exhibited larvicidal activity in the target species (*Culex tritaeniorhynchus*), whereas it was less toxic in non-target bacteria (*Enterobacter ludwigii*). The effects related to the properties of AI-carrier nanopesticides—such as small size or surface area—seem to depend on the cells or organisms being tested.

The nanopesticide class containing inorganic nanoparticles as active ingredients, which is another group evaluated within the scope of this study, includes various metal oxides, such as silicon dioxide (SiO_2_) [35], zinc oxide (ZnO) [36], titanium dioxide (TiO_2_) [37], silver (Ag) [38], copper (Cu) [38,39], and aluminum (Al) [40], which are used against fungi, bacteria, and other plant pathogens. Mondal and Mani [39] reported that a nanoformulation of Cu could suppress bacterial growth on pomegranate at a concentration of 0.2 mg/L, and four times lower than the usual recommended concentration for copper oxychloride (Cu_2_(OH)_3_Cl) (2500–3000 mg/L). This formulation afforded an 8% yield increase as compared to a formulation of copper hydroxide salts currently in use. The authors performed these tests in vitro, and provided no further details about the tested nanopesticide formulation. Therefore, experiments with different nanoformulations should be prioritized as much as possible, and further comparisons should be performed by presenting the properties of nanoformulations in future studies.

The production of Cu-based NPs in 2010 was estimated to reach about 200 tons per year, and this rate has been rapidly increasing [41]. Used as antibacterial, antiviral, and antifungal agents, Cu and Cu-based nanoparticles are also widely employed in cosmetics, paints, coatings, electronics, and pesticides [42,43]. Cu-based nanoparticles and their composites have been shown to cause toxic/genotoxic effects on organisms [44,45,46,47]. On the other hand, some studies in the literature have also determined that CuSO_4_·5H_2_O and copper oxide (CuO) NPs induce no genotoxic effects [46,48]. Cellular Cu homeostasis is tightly regulated under physiological conditions, yet excess Cu is associated with the pathogenesis of hepatic disorders, neurodegenerative changes, and other diseases [49]. Toxicological evidence currently required to evaluate the safety of an active ingredient includes studies involving intravenous and oral absorption, distribution, metabolism and excretion, short-term toxicity (oral, skin, and respiratory), skin and eye irritation, skin sensitization, genotoxicity (both in vitro and in vivo), carcinogenicity, reproductive toxicity, neurotoxicity, and immunotoxicity experiments [18].

Acephate is a type of organophosphorus insecticide that has long been used in the agricultural sector to control chewing and sucking insects such as aphids, fig beetles, leaf yeasts, and straw beetles. Such a large-scale use of acephate not only proves to be costly, but also exerts more toxic effects on agricultural areas [50]. Moreover, its random use has also allowed target species to develop resistance against this insecticide. Nanotechnology-based applications can be a suitable solution to these problems so as to provide a significant reduction in the amount of acetate use, reduce costs, and diminish the toxic effects of acetate while minimizing the resistance in target species. Choudhury et al. [51] reported that another formulation based on polyethylene glycol (PEG)—whose active ingredient is acephate—had greater efficacy against target organisms and lower toxicity than its commercial counterparts. Rajak et al. [52,53] stated that acute sublethal acephate exposure induced glutathione, malondialdehyde, catalase (CAT), superoxide dismutase (SOD), and single-stranded DNA damage in hemocytes (i.e., blood cells) of *D. melanogaster*. In addition, a decrease in locomotor activity [52] and flight movement [54] was observed in *D. melanogaster* as a result of exposure to acephate. The greater efficacy of PEG-based nanoformulations is attributed to the slow release rate and preservation of their relatively unstable active ingredients. However, as with nanoemulsions, the reasons for their lower toxicity in non-target organisms have yet to be clarified.

In inorganic nanoparticles involving an organic active ingredient, mesoporous SiO_2_ is used as a carrier for slow release, or NPs incorporate TiO_2_ into a polymer matrix to catalyze the photodegradation of organic active ingredients [55]. Three new formulations have been proposed for the use of SiO_2_ [56,57] or calcium carbonate (CaCO_3_) [58] as a carrier to allow the slow release of an organic active ingredient. CaCO_3_, the most common inorganic biomineral, exists as three anhydrous crystalline polymorphs (calcite, aragonite, and vaterite) [59]. Due to its low toxicity and slow biodegradability, it is often used as a carrier for drugs and bioactive proteins [60,61]. All three of these nanoformulations have been shown to increase the potency of active ingredients; however, such findings cannot be generalized, due to the very different nature of the active ingredients studied. Both laboratory and field tests have shown that the insecticidal activity of chlorfenapyr-containing SiO_2_ nanoparticles is two times greater than that of the microparticle form of chlorfenapyr [56]. However, this result may be associated with the natural properties of SiO_2_ NPs. Mingming et al. [57] showed that formulations containing SiO_2_ NPs improved the performance of a plant growth regulator as compared to those using pure active ingredients, which could possibly be explained by the slow release of active ingredients (for over 10–20 weeks) that are toxic if applied at high doses. Similarly, the mode of action of validamycin was similar when formulated with CaCO_3_ NPs [58]. Validamycin is a water-soluble antibiotic used to inhibit *Rhizoctonia solani*—a fungal disease agent affecting various soil-borne hosts [62]. Recent research on validamycin shows that it has a larvicidal effect on *D. melanogaster* [54], and impairs larval and pupal development in *A. aegypti* [63].

This study will hopefully encourage scientific efforts towards developing nanopesticides that are less harmful to nature than conventional pesticides. *D. melanogaster* (commonly known as the fruit fly or vinegar fly) (Diptera: Drosophilidae) is a dynamic eukaryotic testing model in genetic and molecular research into human genetic diseases. As an in vivo model organism, *Drosophila* contains approximately 75–77% of genes with homologs in humans [64,65], responsible for diseases such as Parkinson’s, Alzheimer’s, cancer, neurodegenerative disorders, cardiovascular diseases, immune disorders, and intestinal infections [66,67]. Even though there have been several studies into the effects of nanopesticides on target organisms [68,69,70], it is equally crucial to analyze the effects of nanopesticides on non-target species. This project therefore aimed to explore the cytotoxic and genotoxic impacts of different nanopesticide formulations (i.e., a permethrin nanopesticide with nanoemulsion properties, a copper(II) hydroxide [Cu(OH)_2_] nanopesticide as an inorganic NP, an acephate nanopesticide as a polymer-based nanopesticide, a and validamycin nanopesticide as an inorganic nanoparticle associated with an organic active ingredient), as well as microparticle forms of these nanopesticides (i.e., permethrin, copper(II) sulfate pentahydrate [CuSO_4_·5H_2_O], acephate, and validamycin) on *D. melanogaster*—a non-target eukaryotic model organism. The experiments included a *Drosophila* comet assay (for primary DNA damage and oxidative DNA damage in hemocytes and midgut cells), *Drosophila* SMART to detect any somatic mutation and/or recombination activity in wing imaginal disc cells, and measurements of intracellular reactive oxygen species (ROS) in the hemocytes and midgut cells of *Drosophila larvae*, along with assessments of oxidative stress effects (glutathione (GSH) amount), lipid peroxidation product formation (malondialdehyde), levels of nanopesticides crossing the intestinal barrier (and their effects on the intestinal lumen), changes in the expression of genes controlling the integrity of the intestinal barrier and general stress genes, phenotypic changes in different generations (F0, F1, F2, and F3), locomotor behavior (climbing and/or walking), and morphological abnormalities. The selected nanopesticides had not been used in any previous research for this purpose; hence, the determination of the nanotoxic and nanogenotoxic potential of these substances through in vivo and in vitro testing was a rather important step towards establishing the biosafety profiles of these substances. In addition, we also evaluated whether there was a possible correlation between the bulk (i.e., microparticle form) and nanoscale (i.e., different nanopesticide formulation types) forms of these compounds that may influence the extent of their toxicity and genotoxicity. In this respect, this research project successfully fills a knowledge gap in the literature regarding these nanopesticides. The findings derived from our experiments may provide valuable data concerning the cytotoxic and genotoxic potential of nanopesticides widely used in the agricultural sector.

## 2. Results

### 2.1. Characterization of Permethrin Nanopesticides, Cu(OH)_2_ Nanopesticides, Acephate Nanopesticides, and Validamycin Nanopesticides

The nanopesticides used in our study were characterized by TEM, SEM, XRD, HPLC, DLS, and LDV. The findings are presented in the Appendix A (Appendix A). Examples of TEM and SEM images are presented in Appendix A. Particle distributions obtained from TEM images of the tested nanopesticides were examined, and the average particle sizes were found to be 20 ± 5 nm, 15 ± 6 nm, 55 ± 25 nm, and 1177 ± 220 nm, respectively (Appendix A). The average diameters obtained using DLS were 98 ± 10 nm, 105 ± 5 nm, 408 ± 0.00 nm, and 1551 ± 0.00 nm for permethrin nanopesticides, Cu(OH)_2_ nanopesticides, acephate nanopesticides, and validamycin nanopesticides, respectively (Appendix A). The polydispersity index (PDI) values obtained using DLS were 0.677, 0.515, 0.170, and 0.341 for permethrin nanopesticides, Cu(OH)_2_ nanopesticides, acephate nanopesticides, and validamycin nanopesticides, respectively. The PDI values expressing the particle size distribution for the abovementioned nanopesticides were in the range of 0.170–0.677. These ratios show that the dispersion of the nanopesticides in the prepared suspensions was not highly monodisperse; on the contrary, it was polydisperse. Different size-distribution algorithms showed that the PDI values were between 0.05 and 0.7. The calculations used to determine the size and PDI parameters are defined in ISO standard documents (13321: 1996 E and ISO 22412: 2008) [71]. The average zeta potential calculated by the LDV technique was −61 ± 12 mV and −67 ± 9 mV for permethrin nanopesticides and Cu(OH)_2_ nanopesticides, respectively. Greater hydrodynamic diameter and lower zeta potential values indicate the existence of a tendency for nanoparticles to aggregate. To prevent such aggregation, we carried out all experiments by utilizing freshly prepared dispersions through the sonication process. The amount of active permethrin in the synthesized permethrin nanopesticides was determined using the Shimadzu Prominence HPLC System (Japan) and Inertsil ODS-3 C18 column. The measurements were carried out at 40 °C with a 20 µL injection, a 1 mL/min flow rate, and a 90:10 acetonitrile:water mobile phase; the amount of permethrin was determined with a UV detector at 225 nm. To that end, firstly, a calibration chart was created with 20, 40, 60, 80, and 100 ppm standard permethrin solutions (Appendix A).

The crystal structure of Cu(OH)_2_ nanopesticides was characterized by XRD measurements performed according to the powder diffraction technique using the Empyrean X-ray diffractometer (Malvern Panalytical) (Appendix A). Measurements were conducted using Cu Kα irradiation at values of 2θ in the angle range of 10–90° and at a scanning speed of 0.01°. The X-ray diffraction pattern obtained corresponds to the diffraction of Cu(OH)_2_ in the crystalline phase, according to international powder diffraction standards (JCPDS; Joint Committee on Powder Diffraction Standards) [72]. In the diffraction pattern in Appendix A, the peaks at angles of 17.0°, 24.1°, 34.2°, 36.0°, 38.2°, 39.9°, 53.7°, 56.7°, 63.2°, 65.0°, 68.6°, and 74.0° 2θ correspond to reflections from the (240) planes belonging to the crystal phase Cu(OH)2 (020), (021), (002), (111), (041), (130), (132), (061), (113), (200), and (221), respectively (Appendix A).

X-ray diffraction measurements were taken from powder samples taken at different reaction times in order to understand whether there was a change in the crystal structure of the CaCO_3_ particles during the reaction (Appendix A). Appendix A also includes the XRD diffraction pattern of NVal-encoded CaCO_3_ particles containing validamycin on a large scale. When Appendix A is examined, it is clear that the crystal phases of the CaCO_3_ particles are protected during the reaction stages. The X-ray diffraction pattern obtained corresponds to the diffraction of CaCO_3_ in the vaterite phase, according to international powder diffraction standards (JCPDS) [73,74]. In the diffraction pattern in Appendix A, the peaks at angles of 21.3°, 25.3°, 27.4°, 33.1°, 44.2°, 49.4°, 50.3°, and 56.2° 2θ belonging to CaCO_3_ in the crystal phase correspond to reflections from the planes (004), (110), (112), (114), (300), (304), (118), and (224), respectively.

### 2.2. The Endotoxin Levels of Permethrin Nanopesticides, Cu(OH)_2_ Nanopesticides, Acephate Nanopesticides, and Validamycin Nanopesticides

LAL assays were utilized to measure the endotoxin levels (Appendix A). Endotoxin levels at all different doses were found to be below the reference values (0.116667 EU/mL). At the greatest doses (2.5 or 5 mM), the levels of endotoxins were 0.045, 0.048, 0.050, and 0.052 EU/mL, respectively so none of the nanopesticides were contaminated with endotoxins (Appendix A).

### 2.3. Determination of the LC_50_ and Mortality Values of Nanopesticides

The mortality percentages of the control groups exposed to negative control substances and flies exposed to nanopesticides are presented in Appendix A. Concentrations in the range of 0.01–10 mM were examined in terms of LC_50_ values. The LC_50_ value for permethrin was 0.1 mM, while it was 2.5 mM for permethrin nanopesticides, validamycin, and validamycin nanopesticides, and 5 mM for CuSO_4_·5H_2_O, Cu(OH)_2_ nanopesticides, acephate, acephate nanopesticides, and CaCO_3_ (Appendix A). The permethrin nanopesticides and CuSO_4_·5H_2_O at the highest dose (10 mM) significantly increased the lethality in the larval stages (Appendix A). At the highest dose (10 mM), the mortality of the acephate nanopesticides was lower than that of the other three nanopesticides (Appendix A).

### 2.4. Toxicity

Before any testing, the toxicity of the solvent (2% ethanol) and nanocapsules used during the preparation of nanopermethrin (permethrin nanopesticide LC_50_ = 2.5 mM value) was assessed. It had a viability (egg-to-adult survival) of 94 and 93% for the tested concentrations of ethanol and nanocapsules, respectively. The viability of clear distilled water was 100%. The toxic effects of the exposure to the compounds, applied throughout the entire larval development, were specified as variations in the ability to reach the adult stage. When applied at different concentrations from 0.01 to 10 mM throughout the entire larval development, a toxic effect was observed at the applied concentrations of 0.5, 1, 2.5, 5, 7.5, and 10 mM on the ability of permethrin-exposed larvae to reach the adult stage, while no toxic effects were observed at other concentrations. On the other hand, a toxic effect was observed at the applied concentrations of 5, 7.5, and 10 mM upon exposure to permethrin nanopesticides, whereas no toxic effect was observed at other concentrations (Figure 1A). As a result, after the toxicity study for permethrin, the concentration range was determined to be 0.01, 0.03, 0.06, and 0.1 mM in other tests, while it was determined to be 0.01, 0.1, 1, and 2.5 mM for permethrin nanopesticides (Figure 1A).

Although a toxic effect was observed upon exposure to 7.5 and 10 mM CuSO_4_·5H_2_O and Cu(OH)_2_ nanopesticides, impairing the ability to reach the adult stage, no toxic effect was observed at other concentrations. After the toxicity study for both CuSO_4_·5H_2_O and Cu(OH)_2_ nanopesticides, the concentration range was determined to be 0.01, 0.1, 1, and 5 mM in the tests used within the scope of this project (Figure 1B).

The viability rate for PEG-400 was determined to be 94%. This value shows that the capsules used for acephate nanopesticides did not have any significant toxic effects. While a toxic effect was observed at doses of 7.5 and 10 mM after acephate and acephate nanopesticide exposure, no toxic effect was observed at other concentrations. After the toxicity study for both acephate and acephate nanopesticides, the concentration range was determined to be 0.01, 0.1, 1, and 5 mM in the tests used in this project (Figure 1C).

The viability rate for CaCO_3_ was 93%. This value indicates that the nanocapsules used for the validamycin nanopesticides did not have any significant toxic effects. Although a toxic effect was observed at doses of 5, 7.5, and 10 mM following exposure to CaCO_3_, validamycin, and validamycin nanopesticides, no toxic impact was noted for other concentrations. Therefore, the concentration range for these materials was determined to be 0.01, 0.1, 1, and 2.5 mM in the tests. On the other hand, after the toxicity study of CaCO_3_, the concentration range was determined to be 0.01, 0.1, 1, and 5 mM (Figure 1D).

### 2.5. Morphological Alterations

We also looked into the possible effects that might occur during metamorphoses. It should be noted that *Drosophila* flies grow into adults inside the pupae, so they are exposed to compounds from the larval imaginal discs throughout the entire larval development. A thorough morphological analysis of various structures found in adult flies—such as the eyes, mouth, abdomen, legs, and wings—revealed that morphological alterations occurred upon exposure to the highest concentrations of nanopesticides and microparticle forms of these nanopesticides (Figure 2A–E).

Figure 2A illustrates a lateral/dorsal view of an adult fly, showing the different body structures analyzed. In Figure 2B,C, abdominal and wing anomalies, respectively, can be observed in *Drosophila* adults exposed to permethrin (0.1 mM), permethrin nanopesticides (2.5 mM), acephate and acephate nanopesticides (5 mM), and validamycin and validamycin nanopesticides (2.5 mM). As shown in Figure 2D,E, *Drosophila* adults exposed to permethrin (0.1 mM), permethrin nanopesticides (2.5 mM), acephate and acephate nanopesticides (5 mM), and validamycin and validamycin nanopesticides (2.5 mM) were found to have anomalies in the mouth and leg regions. No morphological changes were observed in the groups treated with CuSO_4_·5H_2_O and Cu(OH)_2_ nanopesticides.

### 2.6. Phenotypic Variations

After exposure to different concentrations of compounds, phenotypic variations were examined in F0 (i.e., parents). Then, adult *Drosophila* individuals belonging to the F1, F2, and F3 generations were obtained from those F0 individuals. The F1, F2, and F3 generations contained both normal and defective flies. In the F0, F1, F2, and F3 generations, the highest doses of permethrin (0.1 mM), permethrin nanopesticides (2.5 mM), acephate and acephate nanopesticides (5 mM), and validamycin and validamycin nanopesticides (2.5 mM) caused phenotypic variations in the mouth and leg regions (Figure 3 and Figure 4). On the other hand, no phenotypic variation was observed in the generations exposed to CuSO_4_·5H_2_O and Cu(OH)_2_ nanopesticides.

Each application concentration was performed in three replicates for each generation. When a total of 1500 flies was observed, phenotypic variations observed in the mouths of individuals treated with permethrin (0.1 mM) amounted to 28 ± 1.8% (420 adult individuals) of the total population. However, the rates of phenotypic variation observed in the mouths of individuals treated with permethrin nanopesticides (2.5 mM), acephate and acephate nanopesticides (5 mM), and validamycin and validamycin nanopesticides (2.5 mM) were 30 ± 0.4% (450 adults), 25 ± 1.3% (375 adults), 29 ± 1.5% (435 adults), 26 ± 1.8% (390 adults), and 27 ± 1.6% (405 adults) of the total population, respectively (Figure 3B–F). Figure 3A shows the normal oral phenotype in generations F0, F1, F2, and F3.

As for the phenotypic variations observed in the legs, the rate of those observed in the legs of flies exposed to permethrin (0.1 mM) was 26 ± 0.3% of the total population (390 adults); the other nanopesticides caused variations at the following rates: 29 ± 0.6% (435 adults), 22 ± 1.8% (330 adults), 27 ± 1.3% (405 adults), 25 ± 1.4% (375 adults), and 26 ± 1.7% (390 adults) of the total population, respectively (Figure 4B–F). Figure 4A shows the normal leg phenotype in generations F0, F1, F2, and F3.

### 2.7. Impact of Nanopesticides on Climbing Behavior in Flies

Impaired climbing ability is often considered to be an indicator of seriously damaged locomotor behavior in flies. Significant differences were detected in climbing behavior between the study and control flies after 7-day exposure to the nanopesticides.

The locomotor activity of 2% ethanol used as a permethrin solvent and the nanocapsules (2.5 mM) used during the preparation of permethrin nanopesticides was determined to be 94 ± 3.2% and 91 ± 3.6%, respectively. The concentrations in permethrin application (0.01, 0.03, 0.06, and 0.1 mM) were determined to be 90 ± 4.6%, 86 ± 5.2%, 81 ± 6.3%, and 67 ± 7.4%, respectively, when compared with the control group, while the concentrations used in permethrin nanopesticide application (0.01, 0.1, 1, and 2.5 mM) were determined to be 88 ± 3.8%, 84 ± 4.7%, 80 ± 5.1%, and 69 ± 5.9%, respectively, when compared with the control group (Figure 5A,B).

As compared with the control group, the concentrations (0.01, 0.1, 1, and 5 mM) of CuSO_4_·5H_2_O and Cu(OH)_2_ nanopesticides were 89 ± 4.2%, 84 ± 5.4%, 80 ± 5.1%, and 60 ± 6.3%, respectively. They were determined to be 90 ± 3.5%, 86 ± 5.2%, 82 ± 4.7%, and 79 ± 5.8% in Cu(OH)_2_ nanopesticide application (Figure 5C).

The locomotor activity upon exposure to the nanocapsules (5 mM PEG-400) used during the preparation of the acephate nanopesticide was determined to be 90 ± 3.6%. The concentrations in acephate (0.01, 0.1, 1, and 5 mM) were determined to be 88 ± 4.2%, 85 ± 4.7%, 80 ± 5.4%, and 52 ± 6.7%, respectively, while the concentrations in acephate nanopesticide application (0.01, 0.1, 1, and 5 mM) were determined to be 91 ± 3.3%, 87 ± 4.2%, 83 ± 4.5%, and 59 ± 5.3%, respectively, as compared to the control group (Figure 5D).

The concentrations in CaCO_3_ application (0.01, 0.1, 1, and 5 mM) were determined as to be ± 4.4%, 90 ± 5.1%, 88 ± 4.8%, and 86 ± 5.7%, respectively, as compared to controls (Figure 5E). On the other hand, the locomotor activity in the nanocapsules (2.5 mM CaCO_3_) used during the preparation of the validamycin nanopesticide was determined to be 89 ± 3.9%. When the concentrations (0.01, 0.1, 1, and 2.5 mM) in validamycin and validamycin nanopesticide applications were compared with those of the control group, they were determined to be 87 ± 4.4%, 84 ± 5.1%, 79 ± 5.3%, and 53 ± 5.9% respectively, while in validamycin nanopesticide applications, they were determined to be 89 ± 3.2%, 86 ± 4.6%, 81 ± 4.9%, and 63 ± 5.3%, respectively (Figure 5F).

The highest doses of permethrin (0.1 mM), permethrin nanopesticides (2.5 mM), CuSO_4_·5H_2_O (5 mM), acephate and acephate nanopesticides (5 mM), and validamycin and validamycin nanopesticide (2.5 mM) were found to impair locomotor activity at statistically significant levels.

### 2.8. Oxidative Stress Assay

The highest two concentrations of nanopesticides caused statistically significant changes in glutathione levels in *Drosophila* larvae. On the other hand, there was no significant difference in glutathione levels upon exposure to Cu(OH)_2_ nanopesticides and CaCO_3_ as compared to the control substances (Figure 6A–F).

### 2.9. Lipid Peroxidation Assay

The highest two doses of nanopesticides caused statistically significant changes in the lipid peroxidation formation levels in the larvae. However, Cu(OH)_2_ nanopesticides and CaCO_3_ caused no significant differences in lipid peroxidation formation levels as compared to controls (Figure 7A–F).

### 2.10. Internalization via the Intestinal Barrier

The interactions of the lowest and highest concentrations of nanopesticides and control substances with the *Drosophila* intestinal barrier, effects on the cytoplasmic matrix of intestinal enterocytes of larvae (i.e., epithelial cells of the small intestine), effects on microvilli, internalization into the intestinal lumen, and effects on the lumen, as well as their interactions with cellular components (such as the cytoplasm, vacuoles, and mitochondria) in the lumen, are shown in Figure 8A–E.

In the midgut lumen, nanopesticide particles not yet taken into the enterocytes were located in the vicinity of microvilli (Figure 8A), while the particles taken into the enterocytes coexisted with other cell organelles, with a heterogeneous distribution directed from the apical (upper) pole to the basal (lower) pole (Figure 8A–C). We observed that nanopesticide particle densities in vacuoles increased with the concentration (Figure 8D). The increase in the concentration and duration of nanopesticide exposure caused nanopesticides to be found in all cytoplasmic regions (Figure 8E). With these results, the absorption of nanopesticides by enterocytes was demonstrated in electron-microscopic dimensions.

### 2.11. Reactive Oxygen Species (ROS) in Hemocytes and Midgut Cells

ROS inhibition in hemocytes of third-instar larvae was examined after exposure to nanopesticides and negative control substances (Figure 6A–F). The potential presence of such materials in the hemocytes and midgut cells indicates that they might interact with cells and result in biological effects. As expected, nanopesticide exposure induced higher dose-dependent ROS production in hemocytes and midgut cells, reaching significance at the highest test doses (0.06, 0.1, 0.1, 2.5, and 5 mM), which caused significant oxidative stress levels (Figure 9A–C,E–H). ROS values at the highest dose (2.5 mM) using hemocytes and midgut cells were higher in validamycin (295 and 284%, respectively) than in other compounds, suggesting that validamycin caused greater oxidative stress than the other compounds (Figure 9G). On the other hand, no ROS induction was detected at any concentrations of Cu(OH)_2_ nanopesticide or CaCO_3_ applications (Figure 9D,I, respectively).

### 2.12. Genotoxicity Studies

#### 2.12.1. The Wing-Spot Assay

Comparison of study groups with controls revealed no significant changes for all clone types (i.e., small uniform clones, large uniform clones, twin clones, total *mwh* clones, and total clones). In other words, we determined that nanopesticides and their microparticle forms did not cause mutagenic and/or recombinogenic effects in *Drosophila* SMART assays (Figure 10A–F). When the results from the negative (distilled water) and positive (EMS) control groups were examined, we found that they were in concordance with the results obtained from our previous studies [75,76,77,78]. While flies with normal wings (*mwh/flr^3^*) showed mutation and recombination, only clone types forming due to mutation were observed in serrate-wing flies (*mwh/TM3*) thanks to suppression of recombination by their balancer chromosomes. Therefore, no evaluation was conducted in that regard in serrate-wing flies [76,77,78,79,80].

#### 2.12.2. Comet Assay

Comet assays were carried out on hemocytes and midgut cells of *D. melanogaster* to determine the genotoxic effects of the tested pesticides, with the main parameter being tail intensity, or % DNA tail. In addition, cell viability (toxicity) was determined by trypan blue assay, and the concentration range selection was performed. In order to study non-toxic concentrations via the comet method, values of 70% and above were taken into account for cell viability in the test groups [81]. Cell viability assessments were performed for each concentration studied, and considering the results, cell viability was greater than 70% for each concentration (Figure 11A–I). The highest non-toxic doses of permethrin, permethrin nanopesticides, CuSO_4_·5H_2_O, acephate, acephate nanopesticides, validamycin, and validamycin nanopesticides caused significant single-stranded DNA damage in *Drosophila* hemocytes and midgut cells. On the other hand, there were no significant differences in the levels of single-stranded DNA damage at the concentrations of Cu(OH)_2_ nanopesticides and CaCO_3_ as compared to the control group (Figure 11A–I). The positive control ethyl methane sulfonate (EMS, 4 mM) in the comet assay was compared with distilled water in terms of the tail density (%) parameter, which yielded a statistically significant difference. In other words, the positive control (EMS) induced single-stranded DNA damage (Figure 11A–I).

Furthermore, using the bacterial enzymes Fpg and Endo III allowed us to determine whether the significant impact of the highest doses of pesticides predominantly occurred in oxidized purine bases or pyrimidine bases [82]. The bacterial enzyme application showed that the highest concentrations of pesticides caused oxidative damage to the pyrimidine bases (Figure 11J–M). The net oxidative damage levels in the bacterial enzyme applications are presented in Figure 11J–M. The percentages of net oxidative damage in the bacterial enzyme applications were obtained by subtracting the percentage of tail density obtained via the application of the enzyme solution from the percentage of tail density obtained via the application of the Endo III or Fpg enzyme [77,83,84].

### 2.13. Gene Expression Changes

The percentages of mRNA expression of the *Hsp70*, *Hsp83*, *CAT*, *SOD2*, and *p53* genes were examined as an indicator of stress induced and/or inhibited by exposure to permethrin, permethrin nanopesticides, CuSO_4_·5H_2_O, Cu(OH)_2_ nanopesticides, acephate, acephate nanopesticides, validamycin, and validamycin nanopesticides. There was a significant increase in the expression of the *Hsp70*, *Hsp83*, *CAT*, *SOD2*, and *p53* genes in larvae after exposure to permethrin (0.01, 0.03, 0.06, and 0.1 mM) and permethrin nanopesticides (0.01, 0.1, 1, and 2.5 mM). However, a statistically significant increase (upregulation) for the *Hsp70*, *Hsp83*, *CAT*, *SOD2*, and *p53* genes was observed in larvae exposed to 0.06 and 0.1 mM concentrations of permethrin and 1 and 2.5 mM concentrations of permethrin nanopesticides. The greatest increase for these genes was detected at the highest concentrations of both permethrin (0.1 mM) and permethrin nanopesticides (2.5 mM) (Figure 12A,B). The level of statistical significance was taken as *p* ≤ 0.5 and *p* ≤ 0.01. The data represent the mean ± standard error from three independent experiments.

Exposure to pesticides and nanopesticides, as well as different xenobiotic compounds, can activate different metabolic or physiological pathways according to their mechanism of action. Therefore, changes in the expression of many different genes associated with biological effects that may occur as a result of exposure to these chemicals can be used as markers. In this study, to evaluate the responses of permethrin, permethrin nanopesticides, CuSO_4_·5H_2_O, Cu(OH)_2_ nanopesticides, acephate, acephate nanopesticides, validamycin, and validamycin nanopesticides, the genes (i.e., *Hsp70*, *Hsp83*, *CAT*, *SOD2*, *p53*, and *Ogg1*) and some gene markers (i.e., Duox, Hml, Muc68D, and PPO2) associated with the gut barrier response to xenobiotics were used for the first time. After exposure to permethrin (0.01, 0.03, 0.06, and 0.1 mM), no statistically significant differences were observed in the expression of Duox, Hml, Muc68D, and/or PPO2—gene markers associated with midgut and hemocyte interactions; however, a statistically significant downregulation was observed in larvae exposed to permethrin nanopesticides (0.01, 0.1, 1, and 2.5 mM). On the other hand, a significant decrease in mRNA expression of *Ogg1*—a gene associated with DNA repair—was observed at 0.06 and 0.1 mM doses of permethrin, as well as 1 and 2.5 mM concentrations of permethrin nanopesticides. No statistical differences were observed in the expression of Hsp70, Hsp83, CAT, SOD2, p53, Ogg1, Duox, Hml, Muc68D, or PPO2 for the nanocapsules (2.5 mM) used during the preparation of the permethrin nanopesticides. The level of statistical significance was taken as *p* ≤ 0.01. The data represent the mean ± standard error from three independent experiments (Figure 13A,B).

Concentration-dependent decreases were observed in the expression of the *Hsp70*, *Hsp83*, and *SOD2* genes after exposure to CuSO4.5H_2_O (0.01, 0.1, 1, and 5 mM) and Cu(OH)_2_ nanopesticides (0.01, 0.1, 1, and 5 mM). However, a significant decrease in Hsp70 expression occurred after exposure to 1 and 5 mM concentrations of both CuSO_4_·5H_2_O and Cu(OH)_2_ nanopesticides, in SOD2 after 0.01–5 mM CuSO_4_·5H_2_O, and in Hsp831 only after the 5 mM doses of CuSO_4_·5H_2_O and Cu(OH)_2_ nanopesticides (Figure 12C,D). On the other hand, changes in the expression of the CAT and p53 genes were observed as a result of CuSO_4_·5H_2_O exposure (generally depending on the concentration between 0.01 and 5 mM), and statistically significant results were obtained at 1 and 5 mM concentrations (Figure 12C). When the results obtained as a result of exposure to Cu(OH)_2_ nanopesticides were examined, we observed that there was a concentration-dependent decrease in the expression of the Hsp70, Hsp83, CAT, SOD2, and p53 genes. However, statistically significant decreases were only detected in the expression of the *Hsp70* and *SOD2* genes in larvae exposed to the 1 mM concentration of Cu(OH)2 nanopesticides, while significant decreases were found in the expression of all genes in question at the highest concentration (5 mM) of Cu(OH)_2_ nanopesticides (Figure 12D). The level of statistical significance was taken as *p* ≤ 0.5 and *p* ≤ 0.01. The data represent the mean ± standard error from three independent experiments.

After larval exposure to CuSO_4_·5H_2_O (0.01, 0.1, 1, and 5 mM), no statistical changes were observed in the expression of the Duox, Hml, Muc68D, and PPO2 genes, associated with midgut–hemocyte interaction. In the larvae exposed to 0.1, 1, and 5 mM concentrations of CuSO_4_·5H_2_O, concentration-dependent decreases were observed in the mRNA expression of the *Ogg1* gene when compared to the larvae of the control group, and statistically significant results were found at 1 and 5 mM concentrations of CuSO_4_·5H_2_O (Figure 13C). On the other hand, after exposure to Cu(OH)_2_ nanopesticides (0.01, 0.1, 1, and 5 mM), the expression of Duox, Hml, Muc68D, and PPO2 generally decreased in a concentration-dependent manner. Statistically significant decreases were observed in the mRNA expression of Duox and Hml in larvae exposed to 0.01, 0.1, 1, and 5 mM concentrations of Cu(OH)_2_ nanopesticides, as compared to the larvae of the control group. Significant decreases were observed in the expression of Muc68D and PPO2 in larvae after the highest dose of Cu(OH)_2_ nanopesticides (5 mM). Compared to the control group, in larvae exposed to CuSO_4_·5H_2_O, a concentration-dependent decrease was observed in the mRNA expression of the *Ogg1* gene, while a statistically significant decrease was observed at 1 and 5 mM concentrations. Compared to the control group, an increase was observed in the mRNA expression of the *Ogg1* gene in larvae exposed to 0.01, 0.1, and 1 mM concentrations of Cu(OH)_2_ nanopesticides, while a decrease was observed at the 5 mM concentration. However, these results were not statistically significant (Figure 13D). The level of statistical significance was taken as *p* ≤ 0.5 and *p* ≤ 0.01, and the data represent the mean ± standard error from three independent experiments.

In general, exposure to acephate (0.01, 0.1, 1, and 5 mM) and acephate nanopesticides (0.01, 0.1, 1, and 5 mM) caused concentration-dependent increases in the expression of the *Hsp70*, *Hsp83*, *CAT*, *SOD2*, and *p53* genes analyzed in larvae. A significant increase in the expression of these genes was observed in larvae exposed to 1 and 5 mM concentrations of acephate and acephate nanopesticides (Figure 12E,F). The level of statistical significance was taken as *p* ≤ 0.5 and *p* ≤ 0.01.

No statistically significant changes were observed in the expression of Duox, Hml, Muc68D, and PPO2—gene markers associated with midgut–hemocyte interaction—after exposure to acephate (0.01, 0.1, 1, and 5 mM). However, in larvae exposed to 0.01, 0.1, 1, and 5 mM concentrations of acephate, dose-dependent decreases were observed in the mRNA expression of the *Ogg1* gene when compared to the larvae of the control group, and statistically significant results were found at 1 and 5 mM concentrations of acephate (Figure 13E). On the other hand, dose-dependent reductions were observed in the expression of Duox, Hml, Muc68D, and PPO2 following exposure to acephate nanopesticides (0.01, 0.1, 1, and 5 mM). Significant decreases were observed in the mRNA expression of Duox and Hml in larvae exposed to 0.01, 0.1, 1, and 5 mM doses of acephate nanopesticides. Compared to the control group, in larvae exposed to 0.01, 0.1, 1, and 5 mM concentrations of acephate and acephate nanopesticides, a dose-dependent reduction was observed in the mRNA expression of the *Ogg1* gene, while the 1 and 5 mM concentrations of acephate and the 0.1, 1, and 5 mM doses of acephate nanopesticides produced statistically significant results (Figure 13E,F). No statistical changes were observed in the expression of Hsp70, Hsp83, CAT, SOD2, p53, Ogg1, Duox, Hml, Muc68D, or PPO2 for the nanocapsules (5 mM PEG-400) used during the preparation of the acephate nanopesticides. The level of statistical significance was taken as *p* ≤ 0.5 and *p* ≤ 0.01 (Figure 13E,F).

Exposure to CaCO_3_ (0.01, 0.1, 1, and 5 mM) in the nanocapsules used during the preparation of the validamycin nanopesticides caused no significant changes in the expression of the *Hsp70*, *Hsp83*, *CAT*, *SOD2*, and *p53* genes. However, some dose-dependent changes were observed for the Hsp70 gene in all doses of CaCO_3_, with a statistically significant decrease occurring only at the highest dose (5 mM) of CaCO_3_. In addition, a statistically significant increase in the expression of the *p53* gene was observed at the highest concentration of CaCO_3_ (5 mM) (Figure 12I). On the other hand, no significant changes were detected in the expression of Duox, Hml, Muc68D, PPO2, or Ogg1, which were analyzed in larvae after exposure to CaCO_3_ (0.01, 0.1, 1, and 5 mM). The level of statistical significance was taken as *p* ≤ 0.5 (Figure 13I).

Exposure to validamycin (0.01, 0.1, 1, and 2.5 mM) and validamycin nanopesticides (0.01, 0.1, 1, and 2.5 mM) led to dose-dependent changes in the expression of *Hsp70*, *Hsp83*, *CAT*, *SOD2*, and *p53*. A significant increase in the expression of these genes was noted in larvae exposed to 1 and 2.5 mM concentrations of validamycin and validamycin nanopesticides (Figure 12G,H). The level of statistical significance was taken as *p* ≤ 0.5 and *p* ≤ 0.01. No significant changes occurred in the expression of Duox, Hml, Muc68D, and PPO2 after exposure to validamycin (0.01, 0.1, 1, and 2.5 mM). However, in larvae exposed to 0.01, 0.1, 1, and 2.5 mM concentrations of validamycin, dose-dependent decreases were observed in the mRNA expression of the *Ogg1* gene when compared to the larvae of the control group, and statistically significant results were found at 0.1, 1, and 2.5 mM doses of validamycin (Figure 13G). In the meantime, after exposure to validamycin nanopesticides (0.01, 0.1, 1, and 2.5 mM), the overall expression of Duox, Hml, Muc68D, and PPO2 decreased in a concentration-dependent manner. Statistically significant decreases were observed in the mRNA expression of Duox and Hml in larvae exposed to 0.01, 0.1, 1, and 5 mM concentrations of validamycin nanopesticides compared to the control group larvae, while the expression of the *PPO2* gene was statistically significant in larvae exposed to the two highest concentrations of validamycin nanopesticides (1 and 2.5 mM). Moreover, significant decreases occurred in the expression of the *Muc68D* gene at the highest concentration of validamycin nanopesticides (2.5 mM). Compared to the control group, in larvae exposed to 0.01, 0.1, 1, and 2.5 mM concentrations of validamycin and validamycin nanopesticides, a decrease in the mRNA expression of the *Ogg1* gene was observed in general, in a concentration-dependent manner, while statistically significant results were found at doses of 0.1, 1, and 2.5 mM (Figure 13G,H). No statistical differences were observed in the expression of Hsp70, Hsp83, CAT, SOD2, p53, Ogg1, Duox, Hml, Muc68D, or PPO2 for the nanocapsules (2.5 mM CaCO3) used during the preparation of the validamycin nanopesticides. The level of statistical significance was accepted as *p* ≤ 0.5 and *p* ≤ 0.01 (Figure 13G,H).

## 3. Discussion

In this study, we explored the potential risks associated with various nanopesticide formulations and their microparticle forms (i.e., permethrin, CuSO_4_·5H_2_O, acephate, and validamycin) using a non-target in vivo eukaryotic model organism. To this end, we carried out detailed comparative investigations of potential toxic/genotoxic properties, along with their effects on the gene expression and cellular uptake properties of conventional- and nanoformulations of four different pesticides that are widely used in the agricultural sector. The results obtained in this research endeavor support the advantages of using *D. melanogaster* as a reliable model organism to evaluate the biological properties of different nanopesticide formulations and their microparticle forms.

Previous research involving permethrin indicates that its median lethal concentration (LC_50_) in *D. melanogaster* is 46.47 ppm, while its LC_25_ is 22.5 ppm [31]. In addition, in our previous study, which was performed with the LC_25_ value obtained with permethrin alone and with PBO–permethrin combinations at different rates, no genotoxic effects were detected in either application group in the *Drosophila* SMART assays [32]. The SMART assay results for permethrin and permethrin nanopesticides in this project revealed that neither caused any somatic mutation and/or recombination. Sundaramoorthy et al. [33] stated that permethrin caused toxic effects and chromosome damage in erythrocytes and lymphocytes. Permethrin at microparticle and nanoparticle size has been reported to induce toxic effects on different organisms (e.g., *A. aegypti*, *C. tritaeniorhynchus*, *E. coli*, *E. ludwigii*, *B. subtilis*, *C. sativus*, *Z. mays*, *L. esculentum*, and *A. cepa*), and these effects may vary depending on the cell, organism, and particle size [23,34]. Similarly, our tests with high doses showed significant single-stranded DNA damage, ROS formation, lipid peroxidation product formation, and increased glutathione levels.

The literature review showed that the in vivo toxicity and genotoxicity of CuSO_4_·5H_2_O and copper oxide (CuO) NPs after exposure through ingestion in *D. melanogaster* were evaluated using SMART, comet, and lipid peroxidation assays, reporting that such materials induced single-stranded DNA damage, somatic mutation and recombination, ROS formation, and lipid peroxidation product formation [44,45,47]. Meanwhile, some research detected no genotoxic effects of CuSO_4_·5H_2_O and CuO NPs [45,48]. Our study also showed that Cu(OH)_2_ nanopesticides did not cause any negative biological effects on *D. melanogaster*. In addition, considering the results of studies with CuSO_4_·5H_2_O similar to previous findings in the literature, it was observed that CuSO_4_·5H_2_O induced oxidative damage and genotoxicity at high doses. Morgado et al. [21] found that exposure to different forms of copper hydroxide could induce varying levels of copper sensitivity and bioaccumulation in *T. molitor* and *P. pruinosus*. This nanopesticide generally caused milder effects on both species, and although its toxic impact was highly species-specific, the authors concluded that such nanopesticides could be a good alternative for mitigating the environmental impact, due to their lower toxicity to soil organisms as compared to conventional pesticides.

Choudhury et al. [51] noted that PEG-based acephate could induce certain toxic effects, although it had a lower toxicity than its commercial counterpart. It has been found that acute sublethal acephate exposure causes single-stranded DNA damage and oxidative stress in *D. melanogaster* [52,53]. In this project, statistically significant increases were observed in single-stranded DNA damage, ROS formation, lipid peroxidation product formation, and glutathione levels at high doses of acephate and acephate nanopesticides, although no genotoxic effects were observed.

The long-term effectiveness of validamycin formulated with CaCO_3_ nanoparticles can be attributed to the prolonged release of the active ingredient spanning over 14 days [58]. Validamycin was observed to negatively affect the larval and pupal development in *A. aegypti* [63], and had a larvicidal effect on *D. melanogaster* [54]. The application of CaCO_3_ alone in this project determined that it caused no negative biological effects (i.e., somatic mutation and recombination, single-stranded DNA damage, ROS formation, lipid peroxidation, and oxidative stress formation). However, exposure to validamycin and validamycin nanopesticides caused statistically significant increases in single-stranded DNA damage, ROS formation, lipid peroxidation product formation, and glutathione levels at high concentrations, but no genotoxic effects were detected by SMART assays.

The relevant literature notes that CuO NPs can transfer into *D. melanogaster*’s intestinal epithelial cells and cellular components such as the cytoplasm, vacuoles, and/or microvilli in the intestinal lumen [45,47]. Likewise, we also determined that Cu(OH)_2_ nanopesticides migrated into the intestinal epithelial cells of *D. melanogaster*. Decreases in locomotor activity [52] and flight movement [54] have been detected in *D. melanogaster* as a result of acute sublethal acephate exposure. In line with these findings, we found that the locomotor activity decreased at a statistically significant rate upon exposure to the highest doses (5 mM) of acephate and acephate nanopesticides. Moreover, significant biological effects on adult *Drosophila* flies were observed once they were exposed to permethrin, permethrin nanopesticides, CuSO4.5H_2_O, acephate, acephate nanopesticides, validamycin, and validamycin nanopesticides, including the formation of intracellular ROS in the blood cells, hemocytes, and midgut cells; oxidative stress effects (GSH amount); lipid peroxidation product formation (e.g., malondialdehyde); single-stranded DNA damage; oxidative DNA damage; phenotypic changes in different generations; changes in locomotor activity (e.g., climbing behavior); and morphological disorders. In addition, interactions of nanopesticides with the *Drosophila* intestinal barrier; their effects on the cytoplasmic matrix of the intestinal enterocytes (i.e., epithelial cells of the small intestine), microvilli, and intestinal lumen; and their interactions with the cellular components (cytoplasm, vacuoles, mitochondria, etc.) in the lumen are described in this study.

Exposure to pesticides, nanopesticides, or other xenobiotic compounds can activate different metabolic or physiological pathways according to their mode of action. Therefore, the expression of several gene markers associated with cellular uptake, along with biological effects and changes, may be affected after pesticide/nanopesticide exposure. Cells respond to any external influence in a systematic and coordinated manner at the molecular level via a cascading effect. One of the earliest steps involved in the first response is related to the *Hsp* gene family. Changes in the expression of *Hsp* genes in *D. melanogaster* are quite sensitive indicators for any biological stress [85,86]. The stress-inducible Hsp70 protein is thought to have the potential to be a first-line biomarker of cellular changes thanks to its evolutionary conservation and inducibility by a wide variety of inducers [85]. Furthermore, genes involved in antioxidant defense—such as *CAT* and *SOD2*—play a role in cellular responses to any change in free radical balance. The *Ogg1* and *p53* genes, which are involved in DNA repair and genomic integrity, are safe factors for cell health (77,97). The cell-protective *p53* gene appears to be a crucial center in the cellular genotoxic stress response, acting as a transcription factor to elicit cellular functions, including DNA damage and repair, cell-cycle arrest, and apoptosis. The *p53* gene normally accumulates in the nucleus, and is converted into an active form of DNA binding to control several gene sets to prevent proliferation of DNA-damaged cells [87]. The changes in certain genes after exposure to pesticides and nanopesticides—including general stress genes (*Hsp70* and *Hsp83*), those responsible for antioxidant defense (*CAT* and *SOD2*), DNA repair (*Ogg1*), and *p53* (responsible for genomic integrity), as well as some gene markers (Duox, Hml, Muc68D and PPO2) associated with the intestinal barrier response to xenobiotics—were used for the first time in this study.

A significant increase was observed in the expression of the *SOD2*, *p53*, *Hsp70*, *Hsp83*, and *CAT* genes of the larvae after exposure to permethrin (0.06 and 0.1 mM), permethrin nanopesticides (1 and 2.5 mM), acephate and acephate nanopesticides (1 and 5 mM), and validamycin and validamycin nanopesticides (1 and 2.5 mM). Previous works in the literature show that acute sublethal acephate exposure in *D. melanogaster* could induce oxidative stress (*CAT*, *SOD*, *Hsp70*) in hemocytes [52,53]. These results are consistent with our results obtained with 1 and 5 mM concentrations of acephate and acephate nanopesticides.

Exposure to CuSO_4_·5H_2_O and Cu(OH)_2_ nanopesticides led to significant decreases in the expression of the *Hsp70*, *Hsp83*, and *SOD2* genes after larval exposure to 0.01, 0.1, 1, and/or 5 mM doses of CuSO_4_·5H_2_O and Cu(OH)_2_ nanopesticides. In addition, CuSO_4_·5H_2_O exposure (1 and 5 mM) caused an increase in the expression of the *CAT* and *p53* genes, while Cu(OH)_2_ nanopesticide exposure resulted in significant decreases in the expression of the *Hsp70*, *Hsp83*, *CAT*, *SOD2*, and *p53* genes.

As a result of the CaCO_3_ exposure in the nanocapsules used during the preparation of the validamycin nanopesticides, no statistical differences were observed for the *Hsp83*, *CAT*, and *SOD2* genes at 01, 0.1, 1, and 5 mM doses of CaCO_3_, as compared to the controls. In addition, a statistically significant decrease was observed for the *Hsp70* gene at a 5 mM dose of CaCO_3_, while a significant increase was observed for the *p53* gene.

The level of penetration of pesticides and nanopesticides through the intestinal barrier, along with their effects on the intestinal lumen in the larval stage, was evaluated by looking at their effects on the genes and gene markers (i.e., Duox, Hml, Muc68D, and PPO2) that control the integrity of the intestinal barrier. The expression of Duox, Hml, Muc68D, and/or PPO2, which was analyzed in larvae after their exposure to pesticides and nanopesticides, was compared to that of the control group larvae. While no statistical differences were noted for CaCO_3_ and validamycin in the nanocapsules, significant decreases were detected after exposure to permethrin nanopesticides, Cu(OH)_2_ nanopesticides, acephate nanopesticides, and validamycin nanopesticides.

On the other hand, permethrin at doses of 0.06 and 0.1 mM, permethrin nanopesticides at 1 and 2.5 mM, CuSO4.5H2O at 1 and 5 mM, acephate at 1 and 5 mM, acephate nanopesticides at 0.1, 1 and 5 mM, and validamycin nanopesticides at doses of 0.1, 1 and 2.5 mM all caused a significant decrease in the mRNA expression of the *Ogg1* gene—a gene associated with DNA repair. An increase, with no statistical significance, in the mRNA expression of the *Ogg1* gene was observed in larvae exposed to 0.01, 0.1, and 1 mM concentrations of Cu(OH)_2_ nanopesticides as compared to the control group, whereas a decrease was observed at 5 mM. No significant changes were observed in the expression of Duox, Hml, Muc68D, PPO2, and Ogg1 analyzed in larvae after exposure to CaCO_3_ (0.01, 0.1, 1, and 5 mM). The nanocapsules used for the preparation of the nanopesticides caused no significant changes in the expression of Hsp70, Hsp83, CAT, SOD2, p53, Ogg1, Duox, Hml, Muc68D, or PPO2.

The first physical barrier to protect hosts from xenobiotics and toxins is the peritrophic membrane and its epithelial integrity [88,89]. Its peritrophic matrix is a semipermeable membrane, and its permeability in *Drosophila* is partially controlled by crosslinking of proteins through transglutaminases [90]. Mucin 68D serves a key function as a structural component of the peritrophic membrane, and its gene (*Muc68D*) is expressed only in larvae [91]. Decreased expression of the *Muc68D* gene suggests a lower protective role of the peritrophic membrane.

Peroxidase, or dual oxidase (DUOX), is an enzyme that catalyzes ROS generation by localizing in the plasma membrane and phagosome membrane of the NADPH oxidase family proteins in *Drosophila* [92], and it has been proposed to be a key factor responsible for intestinal defense [93]. In addition to causing an antimicrobial response, DUOX also plays a vital role in intestinal permeability, wound healing, and stem cell regulation [67]. Intestinal permeability increased in *Anopheles* mosquitoes with decreased DUOX expression, due to decreased dityrosine crosslinking of the peritrophic membranes [94]. This is consistent with our results that DUOX expression was significantly inhibited upon exposure to pesticide and/or nanopesticide doses, which would reflect their cellular uptake and affect the luminal microbiota.

Hemolectin (Hml) is a clotting factor in *Drosophila* larvae [95], required to close wounds to prevent hemolymph loss (hemostasis) and prevent the spread of microbes into the hemocoel. Impaired expression of this gene in flies can be induced by any injury [96]. Our results, showing significantly reduced Hml expression, raise the possibility that pesticides and/or nanopesticides could cause tissue injury at the intestinal barrier. Likewise, PPO2 is a prophenoloxidase responsible for rapid melanization at the site of infection or injury to facilitate wound healing or sealing [97]. In this context, low expression of PPO2 after exposure to pesticides and/or nanopesticides in this study demonstrates their deleterious effects on the intestinal barrier. Based on our findings, we can suggest that the genes and gene markers (Duox, Hml, and PPO2) involved in wound healing (among other functions) could be key biomarkers to show lesions in the intestinal barrier caused by pesticides and/or nanopesticides. In this regard, whole-genome analysis is thought to be an appropriate approach to discover the genes/pathways involved in the toxic response to such materials. However, within the scope of this project, we aimed to investigate the role of some candidate genes whose potential role in maintaining epithelial integrity is already known. Our positive findings support this chosen approach, and provide us with an important tool for use in future studies.

When other studies in the literature were reviewed, we saw that a significant decrease was noted for Hsp70 upon exposure to CuO NPs at doses of 0.08, 0.4, 2, and 10 mM, while significant effects were observed only at 0.08 mM after CuSO_4_ treatment. In both CuO NPs and CuSO_4_, copper caused a significant reduction in SOD at almost all doses. CuSO_4_ exposure caused a significant increase in the expression of the *CAT* gene at all tested doses, while the expression of the *CAT* gene following CuO NP exposure caused a statistically significant decrease—especially at 10 mM. In addition, a significant decrease and increase in expression of the *p53* gene was observed after exposure of *Drosophila* larvae to both CuO NPs (10 mM) and CuSO_4_ (2 mM) [45]. In this project, a statistically significant concentration-dependent decrease was detected in the expression of the *Hsp70* and *SOD2* genes, while a significant increase in the expression of the *CAT* and *p53* genes was detected at 0.01, 0.1, 1, and 5 mM concentrations of CuSO_4_·5H_2_O, in accordance with the studies in the literature. Dose-dependent decreases in the expression of Duox, Hml, Muc68D, PPO2, and upd3—the genes associated with midgut–hemocyte interaction—were detected in *Drosophila* larvae exposed to different doses of CuO NPs (0.08, 0.4, 2, and 10 mM). Although no changes occurred in the expression of the *Muc68D* gene, a significant decrease was observed in the expression of Duox, Hml, and Upd3. Furthermore, a statistically significant effect was recorded in the expression of the *PPO2* gene only at the high concentration (10 mM). However, no significant change was noted in the expression of the *Ogg1* gene [45].

Despite there having been plenty of research into the effectiveness of nanopesticides on target organisms [68,69,98], their impact on non-target species still remains surprisingly understudied. EU regulations of the use of nanopesticides in agriculture have recently begun to focus on non-target species. However, no specific test has been developed for nanopesticides so far. The risk assessment of nanopesticides should take into account both the chemical structure and nanoscale forms of the active ingredients (organic, inorganic, or hybrid). Kah et al. [99] emphasized that we still have rather limited data on the effects of active ingredients in traditional pesticides and related nanoformulations—at least not enough to make comparisons between the two product types.

Some previous works have dealt with the toxic effects of nanopesticides on non-target plant species. For instance, Kumar et al. [23] showed that permethrin nanopesticides (particle size measuring 130 nm in diameter) had no effect on seed germination and root length in *Lycopersicum esculentum*, *Cucumis sativus*, and *Zea mays*. Comparable results were reported for an atrazine polymeric nanopesticide (particle size: about 250 nm in diameter) after being tested on non-target maize plants [100]. Certain studies have employed a metabolic approach to understand how plants reprogram their metabolism when exposed to nanopesticides [60,101]. Grillo et al. [18] summarized some physiological and metabolic effects caused by nanopesticides—such as Cu(OH)_2_ NPs, Cu(OH)_2_ nanowires, Cu NPs, and atrazine-loaded polymeric NPs—on different plant species.

Risk assessment of engineered nanomaterials for terrestrial organisms has been carried out over the last several decades, and nanopesticide research has recently begun to gain momentum. The use of several soil organisms is recommended for ecotoxicological models for the risk assessment of chemicals. While adaptations have been proposed for engineered nanomaterials, including some specific applications such as nanobiomaterials [102,103,104,105], there is no specific standardized methodology to study the toxicity of nanopesticides. The model organisms currently employed in the risk assessment of chemicals include annelids (e.g., *Enchytraeus crypticus* [106] and *Eisenia fetida* [107]), nematodes (e.g., *Caenorhabditis elegans* [98]), and arthropods (e.g., *Acanthoscelides obtectus* [70], *Folsomia candida* [108], and *Hypoaspis aculeifer*). Jacques et al. [109] showed that polymeric nanopesticides of atrazine (diameter: about 300 nm) and paraquat (diameter: about 260 nm), as well as lipid nanopesticides of atrazine/simazine (diameter: about 290 nm), are more toxic than non-nanopesticides to a nematode species called *Caenorhabditis elegans*. Firdaus et al. [110] tested two nanoencapsulated bifenthrin formulations in two worm species (*E. fetida* and *Lumbricus terrestris*), discovering that nanoforms caused approximately 50% more bifenthrin accumulation. From a similar perspective, Neves et al. (2019) showed that commercial inorganic nanopesticides (diameter: >1000 nm) had a higher toxicity to *Folsomia candida* as compared to conventional formulations (Cu(OH)_2_ solution). On the other hand, Pascoli et al. [98] found that neem-oil-based nanoparticles (288 nm in diameter) caused no toxic effects on *C. elegans*, whereas the bulk formulation impaired pharyngeal pumping and GST-4 protein expression. Phanse et al. [111] developed positively and negatively charged nanoparticles as a carriers for the delivery of mosquitocidal dsRNA to mosquitoes, and observed their biodistribution in larvae and cells, reporting that positively charged nanoparticles interacted better with the gastrointestinal tract, were more effectively taken up by cells in vitro, and had distribution in the cytosol, while negatively charged nanoparticles were found throughout the metamorphosis of insects, and were mainly found in the head, body, and ovaries of adults.

Risk assessment of engineered nanomaterials for aquatic organisms has been undertaken for quite some time [112,113]. Jenning et al. [114] reviewed the degradation of nanomaterials in aquatic systems, and detailed the available information on ecotoxicity for a range of organisms in both freshwater and marine environments. Although some inorganic nanopesticides were mentioned in this study, little is known about their interaction with living organisms and ecology. The aquatic organisms tested in toxicity research into nanopesticides include fish species (e.g., *Oncorhynchus mykiss* [115], *Channa punctatus* [116], *Prochilodus lineatus* [117], *Danio rerio* [118,119]), crustaceans (e.g., *Daphnia magna* [17], *Daphnia similis* [120], *Leptocheirus plumulosus* [121], and *Ceriodaphnia dubia* [122]), algae (e.g., *Pseudokirchneriella subcapitata* [120], *Chlorella vulgaris* [123], *Closterium* sp. [118], *Lemna valdiviana* [124]), and amphibians (e.g., *Lithobates catesbeianus* [125,126]). Among crustaceans, significant toxic effects were observed after exposure to Cu(OH)_2_ nanopesticides [17]. When they tested such nanopesticides on *D. magna*, the authors detected a decrease in the expression of detoxification-related genes following short-term (24 h) exposure, although the expression of these genes increased significantly after long-term exposure (48 h). On the other hand, it has been determined that the expression of genes related to the reproductive system varies depending on the dose and exposure time. Vignardi et al. [121] reported that both inorganic nanopesticides such as Cu(OH)_2_ (< 1000 nm) and conventional pesticides (e.g., CuCl_2_) caused similar levels of toxicity in *Leptocheirus plumulosus*.

Various studies have looked into the toxicity or insect-repellent effects of nanopesticides on flying target organisms such as *A. aegypti* [118], *C. tritaeniorhynchus* [118], *C. pipiens* [127], *C. quinquefasciatus* [128], *Spodoptera littoralis* [129], and *Blattella germanica* (German cockroach) [130]. However, more research is needed to assess the risk of nanopesticides on non-target air organisms—particularly those that feed on agricultural products (e.g., honeybees and other vital pollinators). These species are responsible for the pollination of 35% of global farmland, and support the production of 87 of the leading food crops worldwide, according to the Food and Agriculture Organization (FAO) [131]. A significant loss of crop yield may occur indirectly due to a decline in the population of pollinators.

## 4. Methods

### 4.1. Chemicals

Calcium chloride (CaCl_2_, 99.9%), sodium carbonate (Na_2_CO_3,_ 99.9%), N-butanol (C_4_H_9_OH, 99.9%), cyclohexane (C_6_H_12_, 99.9%), methanol (CH_3_OH, 99.9%), ethanol (CH_3_CH_2_OH, 99.9%), acetonitrile (CH_3_CN, 99.9%), validamycin (C_20_H_35_NO_13_, 73.7%), dichloromethane (CH_2_Cl_2_, 99.9%), polyethylene glycol (PEG) (H(OCH_2_CH_2_)_n_OH, ≥98), n-butyl acetate (C_16_H_12_O_2_, 99.9%), sec-butyl alcohol (C_4_H_10_O, 99%), copper(II) sulfate pentahydrate (CuSO_4_·5H_2_O), and calcium carbonate (CaCO_3_) were purchased from Merck (Darmstadt, Germany). Ethyl methanesulfonate (EMS, CAS No. 62-50-0), hydrogen peroxide (H_2_O_2_, CAS No. 7722-84-1), acephate (C_4_H_10_NO_3_PS, ≥98), permethrin (C_21_H_20_Cl_2_O_3_, ≥90), hexadecyl trimethyl ammonium bromide (C_19_H_42_BrN, ≥98), phosphatidylcholine (from soybean lecithin) (C_42_H_80_NO_8_P, 40%), and ammonium glycyrrhizate (C_42_H_62_O_16_.NH_3_, ≥70) were purchased from Sigma Chemical Co. (St. Louis, MO, USA). Copper hydroxide (Cu(OH)_2_) particles were purchased from DuPont as a commercial biocide (Kocide^®^ 3000). For negative control groups, we used sterile distilled water, ethanol (2%), nanocapsules (2.5 mM), nanocapsules (5 mM PEG-400), and nanocapsules (2.5 mM CaCO_3_) for the preparation of the CuSO_4_·5H_2_O and Cu(OH)_2_ nanopesticides, permethrin, permethrin nanopesticides, acephate nanopesticides, and validamycin nanopesticides, respectively.

### 4.2. Synthesis of Permethrin Nanopesticides, Cu(OH)_2_ Nanopesticides, Acephate Nanopesticides, and Validamycin Nanopesticides

Permethrin (C_21_H_20_Cl_2_O_3_) is a chemical compound characterized by rather low water solubility. Therefore, we used the method recommended in the literature to prepare nanopermethrin in a nanometric powder form that could be easily dispersed in water [33]. For this, 1 g of commercial permethrin (at 95% purity) was dissolved in 3.5 g of n-butyl acetate (nBuAC) at 25 °C. Then, 2.25 g of secondary butyl alcohol (sec-BuOH) was added to this solution and mixed at 1200 rpm for 10 min. The organic phase created by adding 3.214 g of soybean lecithin containing 70% soybean phosphatidylcholine to this mixture was mixed at 1200 rpm for 30 min. Afterwards, an aqueous solution of 2.125 g of sucrose and 11.625 g of water was added to this organic phase at a constant temperature of 25 °C, at a rate of about 1 drop per 5 s. The oil/water microemulsion system formed after the addition of the aqueous phase continued to be mixed at 1200 rpm for 1 h. Finally, the obtained isotropic mixture was frozen at −80 °C for 16 h and then dried in a lyophilizer (DW-3 Lyophilizer, Heto-Drywinner) for 24 h.

A commercial biocide (Kocide^®^ 3000, produced by DuPont) was purchased and used for the Cu(OH)_2_ nanopesticide whose toxicity was tested in this study [132,133,134]. Detailed physicochemical properties of Kocide 3000 have been reported in previous studies [132,133]. First, the method to prepare stable dispersions of the DuPont Kocide^®^ 3000 Cu(OH)_2_ nanopesticide containing 35% Cu^2+^ equivalent was optimized. To that end, 1 mg/mL dispersions were prepared after the greenish-blue-colored Cu(OH)_2_ nanopesticide was thoroughly ground in a mortar. As dispersion preparation methods, mixing with a magnetic stirrer at 1200 rpm, holding in an ultrasonic bath, and combining these two methods were systematically tried. The particle sizes and size distributions of the dispersions obtained for each trial were characterized by dynamic light scattering (DLS) measurements on a Malvern Zetasizer ZS device.

Nanoacephate was prepared by encapsulating acephate (C_4_H_10_NO_3_PS) with the hydrophilic polymer PEG [51]. First, 10 mL of a 9:1 PEG-400:water mixture was prepared. While this mixture was stirred at 1200 rpm, 0.1% DCM–acephate solution prepared by dissolving 10 mg of acephate in 10 mL of dichloromethane (DCM) was added to this mixture at 45 °C for about 20 min, and then it was stirred at 1200 rpm at 45 °C for a total of 4 h. The organic solvent residue in the mixture containing acephate encapsulated with PEG-400 was removed by means of a vacuum evaporator. In order to measure the hydrodynamic diameter of the synthesized acephate nanopesticide, 0.5 mL of the acephate nanopesticide solution was mixed with 1.5 mL of ultrapure water, and DLS analyses were performed. To achieve the desired size distribution, we tried several doses containing DCM–acephate at different concentrations and varying ratios.

Synthesis of nano-CaCO_3_ particles containing validamycin (C_20_H_35_NO_13_) was performed by utilizing the reaction of calcium chloride and sodium carbonate via the reverse-phase microemulsion method in a solution medium containing validamycin [58]. The experiments were initially carried out with a surfactant/water (s/w) ratio of 1:20. First, 0.3 g of CTAB, 0.8 g of n-butyl alcohol, and 1.4 g of cyclohexane were mixed at 1000 rpm at room temperature. Then, 100 µL of 1 M calcium chloride was added to this mixture, and it was dried for 1 min. After mixing at 1000 rpm for 5 min, 100 µL of ultrapure water was added, and the mixture was stirred once more at 1000 rpm for 5 min. Thus, by adding 100 µL of 1 M sodium carbonate to the microemulsion system, CaCO_3_ particles were precipitated. The synthesized CaCO_3_ particles were dried at 6000 rpm for 15 min. After separation by centrifugation, they were washed twice with 2 mL of deionized water and methanol, and then dried in a vacuum oven at 313 K for 18 h. The hydrodynamic diameters of the synthesized CaCO_3_ particles were determined by DLS measurements, while their size and morphology were characterized by transmission electron microscopy (TEM) and scanning electron microscopy (SEM) measurements. The changes in the crystal structure of the CaCO_3_ particles were investigated by XRD measurements performed both during the synthesis stages and after validamycin loading.

### 4.3. Characterization and Dispersion of Permethrin Nanopesticides, Cu(OH)_2_ Nanopesticides, Acephate Nanopesticides, and Validamycin Nanopesticides

The characterization of the nanopesticides was achieved through analyses TEM (LEO 906 E TEM by ZEISS) (Austin, Texas, USA), SEM (LEO 1430 by ZEISS) (Hillsboro, Oregon, USA), DLS (Worcestershire, UK), and laser Doppler ve-locimetry (LDV) (Malvern Zetasizer Nano-ZS ZEN 3600) (Worcestershire, UK). A HyperCOOL HC3110 lyophilizer was used in the lyophilization processes. The crystal structure of the CaCO_3_ particles synthesized during the preparation of the Cu(OH)_2_ and validamycin nanopesticides was characterized by X-ray diffraction (XRD) measurements performed on powder samples using the Empyrean X-ray diffractometer from PANalytical (Almelo, the Netherlands). Measurements were performed through Cu Kα irradiation at 2θ in the angle range of 10–90° and at a scanning speed of 0.01°. The permethrin content in the prepared permethrin nanopesticide was characterized using a Shimadzu Prominence High-Performance Liquid Chromatography (HPLC) system with a UV detector at 225 nm and an Inertsil ODS-3 C18 column. Prior to dispersion, a pre-wetting procedure was applied to the permethrin nanopesticides, Cu(OH)_2_ nanopesticides, acephate nanopesticides, and validamycin nanopesticides in 0.5% ethanol, after which the nanopesticides were dispersed in 0.05% BSA in Milli-Q water in order to enhance their workability. Then, the permethrin nanopesticides, Cu(OH)_2_ nanopesticides, acephate nanopesticides, and validamycin nanopesticides were exposed to ultrasonic vibration at 20 kHz for 30 min using a Branson Digital Sonifier system (S-250D) (Danbury, CT, USA) to obtain a stock dispersion of 2.56 mg/mL, in line with previously described protocols [135,136].

### 4.4. Endotoxin Assay

The endotoxin content was measured by chromogenic Limulus amebocyte lysate (LAL) assay (Lonza (QCL-1000TM), Inc., Walkersville, MD, USA), as per the protocol described in the manual and in previous studies [75,137,138,139,140,141]. Pyrogenic material was cleared from the test tubes by heating them at 200 °C. The standard amount (50 mL) of test sample was placed into the wells and heated in 96-well plates at 37 °C. A minimum of three wells were used for each sample. A standard curve was created over the concentration range 0.116667–1.008 EU/mL and compared to the reference range (*Escherichia coli* E50-640) for each assay. Endotoxin standards and diluted samples were assessed for endotoxin testing in pyrogen-free microplates (Costar No. 3596; Corning, Inc., Corning, NY, USA) using a BioTek Synergy 2 microplate reader (BioTek, Winooski, VT, USA) at 37 °C. Absorbance was measured at 405–410 nm. We utilized commercially available control endotoxins (lipopolysaccharide (LPS); 0.5 EU/mL) and LAL water as positive and negative controls, respectively. The percentage of recovery spike values was calculated as follows:(1)$$\mathrm{Recovery}\;\mathrm{of}\;\mathrm{spike}\;\mathrm{value}\;(\%)=\frac{a-b}{c}\times \;100\;$$
where a represents the amount of endotoxin in a spiked sample, b is the amount of endotoxin in the sample, and c represents the amount of added endotoxin. The recovery spike values were calculated as 102.2, 104.2, 101.8, and 102.8% for permethrin nanopesticides (0.01, 0.1, 1, and 2.5 mM, respectively); 102, 106.6, 104.8, and 117.2% for Cu(OH)_2_ nanopesticides (0.01, 0.1, 1, and 5 mM, respectively); 100.6, 107.4, 105.4, and 108.2% for acephate nanopesticides (0.01, 0.1, 1, and 5 mM, respectively); and 105.2, 101.8, 108.4, and 113.2% for validamycin nanopesticides (0.01, 0.1, 1, and 2.5 mM, respectively).

### 4.5. Determination of 50% Lethal Concentration (LC_50_) and Mortality Values

To calculate the LC_50_ values, a total of 25 third-instar *Drosophila* larvae at 72 h were randomly chosen and exposed to different doses of permethrin, permethrin nanopesticides, CuSO_4_·5H_2_O, Cu(OH)_2_ nanopesticides, acephate, acephate nanopesticides, CaCO_3_, validamycin, and validamycin nanopesticides through ingestion (4.5 g of dry *Drosophila* Instant Medium (Carolina Biological Supply Company, Burlington, NC, USA) hydrated with 10 mL of the freshly prepared test solutions). We then recorded the numbers of emerging flies 10 days after exposure, and the larva-killing capacity of the materials was recorded as the percentage of larvae unable to grow into adults. Four replications with 25 larvae were conducted for each treatment. This testing was repeated 3 times on the subsequent days.

The nanopesticides were exposed to the pots 24 h before the testing, and the experiments were performed at room temperature with 50% relative humidity [142,143,144]. Adult *Drosophila* flies were anesthetized with diethyl ether before testing. Then, they were placed into the pots, where they were exposed to insecticides. To allow air transfer, the pots were covered with clean muslin and rubber at the top. After 15 min, the flies were transferred to clean pots. Wet cotton was put on the muslins to maintain humidity. 24 h later, the numbers and rates of the dead flies were recorded.

### 4.6. D. melanogaster Strains, Exposure, Toxicity, and Morphological Alterations

*Drosophila* larvae and adults were cultured at 25 ± 1 °C (in a *12*:*12* h *light*:*dark* cycle) with 60% humidity on a food culture consisting of cornmeal, sugar, yeast, agar, propionic acid, and nipagin. Three different mutant strains were used in the study: wild Canton-S, *flare-3*, and multiple wing hairs. In the wing SMART assay, two distinct fly strains were used: *flare-3* with genetic constitution *flr^3^/In (3LR) TM3*, *Bd^s^* and multiple wing hairs with *mwh/mwh* genetic constitution. The study by Lindsley and Zimm [145] provides more data on the descriptions and genetic markers of these phenotypes. In other experiments, we used the wild Canton-S strain only. In an attempt to assess the toxic potential of permethrin, permethrin nanopesticides, CuSO_4_·5H_2_O, Cu(OH)_2_ nanopesticides, acephate, acephate nanopesticides, CaCO_3_, validamycin, and validamycin nanopesticides, we measured the egg-to-adult survival rates in flies. The adult Canton-S flies were placed in darkened bottles with food culture, facilitating egg collection, which was performed every 8 h. Each test sample containing 50 eggs was transferred to plastic vials, into which we placed 4 g of instant food culture specifically designed for *Drosophila* (Carolina Biological Supply Co., Burlington, NC, USA). The food culture had been saturated with 10 mL of various concentrations of permethrin nanopesticides (0.01, 0.1, 1, and 2.5 mM), permethrin (0.01, 0.03, 0.06, and 0.1 mM), Cu(OH)_2_ nanopesticides and CuSO_4_·5H_2_O (0.01, 0.1, 1, and 5 mM), acephate nanopesticides and acephate (0.01, 0.1, 1, and 5 mM) and validamycin nanopesticides and validamycin (0.01, 0.1, 1, and 2.5 mM) dispersions, and the final concentrations of food culture for permethrin nanopesticides (0.001, 0.011, 0.112, and 0.280), permethrin (0.001, 0.003, 0.007, and 0.011), Cu(OH)_2_ nanopesticides (0.0002, 0.002, 0.018, and 0.091), CuSO_4_·5H_2_O (0.006, 0.064, 0.642, and 3.210), acephate nanopesticides (0.001, 0.005, 0.052, and 0.262), acephate (0.001, 0.005, 0.052, and 0.262), validamycin nanopesticides (0.0004, 0.004, 0.042, and 0.105), validamycin (0.0004, 0.004, 0.042, and 0.105), and CaCO_3_ (0.003, 0.026, 0.257, and 1.287) are indicated as mg/g.

The pesticide doses are expressed as mM throughout the text. Such doses were designated in accordance with previous works in the relevant literature [6,11,12,13,14,15,16,23,31,32,33,34,39,51,55,58,134,146,147].

The ideal EMS dose (1 mM) to cause effective mutagenesis was based on the evidence from our previous works [75,148]. We then ran preliminary tests to confirm the toxic potential of the nanopesticides. The various doses of permethrin nanopesticides, Cu(OH)_2_ nanopesticides, acephate nanopesticides, and validamycin nanopesticides were designated so that the highest dose should not exceed 5 mM. For each dose, we utilized a total of five different replicate samples. The adult flies that survived the pesticide exposure were then collected to determine their counts in order that the survival rate could be calculated. In total, 100 surviving adult flies per treatment were examined for any developmental changes under stereomicroscopy (SLX-2 STEREOZOOM) to assess the impact of exposure during the egg-to-larva phase, and then the morphological changes observed across various body parts—such as the head, thorax, wings, legs, and abdominal area—were recorded in detail [75].

### 4.7. Phenotypic Variations

The F0 (parent), F1, F2, and F3 generation flies in the control groups, along with those in the study groups exposed to test chemicals, were meticulously observed under a stereomicroscope to record any possible phenotypic variations in the head, thorax, eyes, mouth, wings, legs, or abdominal region. In this process, we followed previously published protocols in the literature [147,149,150,151]. A total of 1500 randomly chosen flies (500 flies for each of 3 replications) was analyzed for exposure to each dose of the compounds.

### 4.8. Climbing Assay

In accordance with the procedures proposed in the previous work in the literature, the locomotor behavior of the flies was measured through climbing assays [75,150,151,152,153]. For this experiment, 10 flies from both the control and study groups were separately transferred to vials, where they were acclimatized for 15 min at room temperature. The vials were gently tapped to send the flies down to the bottom, and then their climbing ability was measured by recording the number of flies managing to climb above the 7 cm mark within 10 s in each group. This climbing assay was repeated 10 times for each group after each treatment.

### 4.9. Oxidative Stress Assay

Oxidative stress assays were performed by measuring glutathione (GSH) levels. The methods applied were mainly as follows: 40 larvae were used for each concentration in the control and study groups, and 5 repetitive tests were carried out in each concentration group. After the cuticle was removed, it was homogenized in 1 mL of cold homogenizing solution (0.15 M phosphate solution containing 0.15 M KCl, pH = 7.4) in order to obtain 10% tissue homogenate from the internal tissues of the larvae belonging to the control and nanopesticide-exposed groups. Then, the supernatant to be used in the experiment was obtained by centrifuging at 9000–10,000× *g*. The GSH levels in the exposed larvae were measured using Ellman’s reagent [154,155]. The test mixture was formed from 0.2 M phosphate solution (pH = 8), 0.01% 5.5’-dithiobis-2-nitro benzoic acid (DTNB), and larval homogenate, and the reaction was monitored at 412 nm. The GSH levels in the larvae were expressed in nmol/mg protein.

### 4.10. Lipid Peroxidation Assay

The lipid peroxidation tests were carried out by measuring the amounts of malondialdehyde in the control and study groups. Third-instar *Drosophila* larvae (72 ± 4 h) were exposed to the compounds used as negative controls, nanopesticides, and different concentrations of microparticle forms of the nanopesticides for approximately 24 h. The lipid peroxidation or thiobarbituric-acid-reactive species (TBARS) assay was performed in accordance with the method modified by Carmona et al. [44] and Tironi et al. [156]. The applied methods were as follows: *Drosophila* larvae (each weighing about 0.5 g) were chosen and separately exposed to the control substances, nanopesticides, and microparticle forms of the nanopesticides. The experiments were carried out with two independent tests and three replications in each independent test. The larvae in the control and exposed groups were homogenized with 0.5% *w/v* TCA (trichloroacetic acid). The larval homogenates were kept on ice for 30 min and then filtered. Then, 0.5 mL of the filtered tissue homogenates and 0.5 mL of 0.5% *w/v* TBA (2-thiobarbituric acid) solution were mixed and incubated at 70 °C for 30 min. Then, the absorbance values were measured at 532 nm on a spectrophotometer. The TBARS levels were converted to MDA, and the molar consumption coefficient (1.56 × 105 M^−1^) was used to achieve results displayed as mg of MDA/kg of sample.

### 4.11. Internalization via the Intestinal Barrier

Internalization of nanopesticides within the intestinal barrier was assessed in 3rd-instar *Drosophila* larvae (exposed for four days), which were collected, washed carefully, dissected in phosphate buffer (PB; 0.1 M, pH 7.4), and manipulated in line with our previously described protocol [88,157,158,159]. Firstly, 0.15 M (1X) PBS (137 mM NaCl (sodium chloride), 2.7 mM KCl (potassium chloride), 4.3 mM Na_2_HPO_4_ (disodium hydrogen phosphate), 1.4 mM KH_2_PO_4_ (potassium dihydrogen phosphate)) was prepared, and then 0.15 M PBS was diluted with distilled water at a ratio of 2:1 to obtain 0.1 M PBS. Subsequently, 72 ± 4 h old *Drosophila* larvae were exposed to the lowest and highest concentrations of nanopesticides (0.01 mM and 2.5 mM for permethrin nanopesticides, 0.01 mM and 5 mM for Cu(OH)_2_ nanopesticides, 0.01 mM and 5 mM for acephate nanopesticides, and 0.01 mM and 2.5 mM for validamycin nanopesticides, respectively). After 24 h of exposure, midguts from 96 ± 4 h old larvae from the control and treatment groups were dissected in 0.1 M phosphate-buffered saline (PBS: 0.1 M, pH: 7.4). Then, a fixative solution containing 4% paraformaldehyde and 1% glutaraldehyde was obtained by adding 4 g of paraformaldehyde and 0.1 mL of glutaraldehyde to 100 mL of 0.15 M PBS. Midguts isolated from the larvae were fixed in a fixative solution for 2 h at +4 °C. Midgut dissection performed in *Drosophila* specimens is shown in the literature [158,159,160]. The following methods were used to examine the isolated midguts through TEM: For fixation, the samples were kept in 4% glutaraldehyde (prepared in 0.1 M Sorensen phosphate buffer) at +4 °C for 2 h. They were washed for 10 min (3 repetitions) on a rotator in 0.1 M Sorensen phosphate buffer at room temperature. Osmium tetroxide (OsO_4_) fixative prepared in 1% Sorensen phosphate buffer (prepared with 1% OsO_4_ and 0.8% potassium hexacyanoferrate II) was fixed on the rotator at room temperature for 2 h. Washing was performed with 0.1 M Sorensen phosphate buffer at room temperature in 3 repetitions for 10 min on the rotator. The bottles were changed in the last wash. In the dehydration process, they were kept in 50% and 70% ethanol through 3 repetitions at +4 °C for 2 × 15 min. En bloc staining, which is an electron microscopy staining technique, was performed at +4 °C for 1 h in a 1% uranyl acetate solution prepared with 75% ethanol. As a continuation of the dehydration process, it was kept in 80%, 90%, and 100% ethanol for 10 min at +4 °C for 3 repetitions. Before the embedding of the samples, the tissues were kept on the rotator with propylene oxide for 10 min and 4 h at room temperature with propylene oxide + araldite master mix + 2% accelerator benzyldimethyl amine (BDMA) (1:1) as an intermediate solution for 2 repetitions. Before the embedding process, the pieces were removed from the bottles with the help of toothpicks, and the excess araldite was removed by rolling on the filter paper. Pieces were taken from this mixture in a separate bottle, rotated at room temperature for 1 night, and embedded in the mixture (1st araldite). After embedding, it was kept at 60 °C for 48 h. Once it had become a block, Cu was placed on grids, and thick and thin sections were obtained. After the samples were examined via TEM, the cellular uptake of nanopesticides through the intestinal barrier into the intestinal lumen of *Drosophila* larvae was evaluated using the images.

### 4.12. Intracellular Oxidative Stress (ROS) Detection

Intracellular ROS generation was assessed by dichloro-dihydro-fluorescein diacetate (DCFH-DA) screening in hemocytes and midgut cells from *Drosophila* larvae once they had been exposed to the chemicals [161]; the procedure is described in our previously published papers [88,138,157,158,159]. To summarize, after their collection from the flies, the hemocytes and midgut cells were exposed to 5 mM DCFH-DA for 30 min at 24 °C. The fluorescence was measured using a fluorescent microscope with an excitation of 485 nm and an emission of 530 nm (green filter). The ImageJ software package was used to quantitatively assess the fluorescent images from both controls and treated larvae [158,162].

### 4.13. Comet Assay

DNA breakage was assessed by comet assays, where the hemocytes collected from the *Drosophila* larvae were used as target cells. The procedures described in previously published works in the literature were followed at every stage [75,138,159,161,163,164]. Third-instar larvae (72 ± 4 h old) were transferred into plastic vials previously filled with 4 g of *Drosophila* instant food culture that was pre-wetted with the following concentrations: control (distilled water) at 0, and study groups at different concentrations of permethrin nanopesticides (0.01, 0.1, 1, and 2.5 mM), permethrin (0.01, 0.03, 0.06, and 0.1 mM), Cu(OH)_2_ nanopesticides and CuSO_4_·5H_2_O (0.01, 0.1, 1, and 5 mM), acephate nanopesticides and acephate (0.01, 0.1, 1, and 5 mM), and validamycin nanopesticides and validamycin (0.01, 0.1, 1, and 2.5 mM) dispersions, for 24 ± 2 h. Clean distilled water was used as a negative control [76], while EMS (4 mM) [159,161] was used as positive control.

The comet assay can assess single-stranded DNA damage as well as oxidative damage to DNA using some bacterial enzymes. Therefore, using the bacterial enzymes endonuclease III (Endo III) and formamidopyrimidine DNA glycosylase (Fpg) isolated from *Escherichia coli* bacteria, the levels of oxidized purine and pyrimidine bases can be determined by modifying the comet assay. Endo III shows the number of specifically oxidized pyrimidine bases, while Fpg shows the number of specifically oxidized purine bases [77,82,83,84]. The chemical applications involved the highest concentrations of the tested nanopesticides in hemocytes and midgut cells isolated from *Drosophila* larvae under in vivo conditions. Enzyme application was performed by selecting certain enzymes to assess whether the observed DNA damage was caused by purine bases or pyrimidine bases.

### 4.14. Gene Expression Changes

We also examined whether there were changes in the expression of certain genes. The first set of genes was associated with general stress: heat-shock protein 70 (*Hsp70*, NM_169441.2), heat-shock protein 83 (*Hsp83*, NM_001274433.1), catalase (*CAT*, NM_080483.3), superoxide dismutase (*SOD*, NM_057577.3), *p53* (NM_206544.2), and *Ogg1* (NM_132271.5). The second set of genes was associated with the intestinal barrier’s response to physical stress: dual oxidase (*Duox*, NM_001273039.1), hemolectin (*Hml*, NM_079336.3), mucin 68D (*Muc68D*, NM_140247.3), and prophenoloxidase 2 (*PPO2*, NM_136599.4). The expression of the selected genes in treated and untreated larvae was detected through homogenized groups of 30 third-instar larvae (50 mg) in TRIzol^®^ Reagent (Invitrogen, Carlsbad, CA, USA). RNase-free DNase I (DNA-free^TM^ kit; Ambion, Paisley, UK) was used to remove DNA contamination. The quantity of mRNA in each sample was determined by means of a NanoDrop device. cDNA was synthesized through the use of 1 mg of total RNA with the Transcriptor First-Strand cDNA Synthesis Kit (Roche, Basel, Switzerland), and stored at 20 °C. The resulting cDNA was subjected to real-time RT-PCR analysis on a Light Cycler 480 (Roche, Basel, Switzerland) to determine the relative expression of the selected genes by using β-actin as the housekeeping control. The reaction conditions for all genes were as follows: one cycle of pre-incubation for 5 min at 95 °C, and amplification was repeated 45 times (10 s at 95 °C, 15 s at 61 °C, 25 s at 72 °C). The data were analyzed using the StepOne^TM^ software v 2.2.2. The mean cycle threshold (Ct) value for the targeted gene was used to calculate the relative expression with the relative quantification (RQ) value and formula: RQ = 2^−ΔCT^ × 100, where ΔCT = CT of target gene − CT of an endogenous housekeeping gene. Evaluation of 2^−ΔCT^ reveals the fold change in gene expression, normalized to the internal control (β-actin), enabling the comparison of the larvae treated with different substances [88,138,157,158].

### 4.15. Statistical Analysis

The frequency differences between each type of spot in exposed flies and negative controls were evaluated by means of a conditional binomial test, tabulated in detail by Kastenbaum and Bowman [165], with a significance level of α = β = 0.05. Data from the tests were analyzed by the Mann–Whitney *U*-test to exclude false positive and negative values [166,167]; *p*-values equal to or less than 0.05 were accepted as statistically significant.

The normality of variance was analyzed using the Kolmogorov–Smirnov and Shapiro–Wilk tests, and homogeneity of variance was evaluated by Levene’s test. Data with normal distribution and equal variance were analyzed using Student’s *t*-test (endotoxin assay, glutathione (GSH) assay, comet assay, and gene expression changes) and one-way ANOVA (climbing assay and lipid peroxidation assay) using SigmaPlot version 11.0 (SPSS, Chicago, IL). Mean mortality values for 24 h were calculated and compared through Duncan’s multiple range test (*p* ≤ 0.05), and the LC_50_ values were calculated using a statistical analysis tool described in [168]. Data with unequal variance or skewed distribution (i.e., viability/toxicity and ROS production) were assessed by the non-parametric Mann–Whitney *U*-test. All data from the experiments are presented as the means of two independent experiments, including duplicates of each one, unless stated otherwise. The values are presented as arithmetic means ± standard error.

## 5. Conclusions

The widespread use of nanopesticides in agricultural areas has recently triggered a scientific interest in determining the merits and drawbacks of nanopesticides. In that regard, this study stands out as an original research endeavor, as it explores the effects of different types of nanopesticides and microparticle forms of pesticides through assays for toxicity, genotoxicity, phenotypic and behavioral changes, biochemical impacts, cellular uptake, and gene expression in *D. melanogaster*—a eukaryotic in vivo model organism. This completed project provides a useful framework in determining the priorities for future in vivo research towards stronger risk analysis of nanopesticides used in agricultural areas all over the world. The main conclusions can be summarized as follows:(1)Characterization of nanoparticles for size and diameter should be specified in detail by TEM and SEM imaging techniques, along with XRD and HPLC.(2)Endotoxin levels at all different doses of test chemicals were below the detectable limit level (0.116667 EU/mL), and these nanopesticides were not contaminated with endotoxins.(3)Lower doses of the tested nanopesticides showed no hazardous effects on the ability of fruit flies to reach the adult stage.(4)All nanopesticides other than copper-based pesticides caused morphological changes in the abdomen, wing, mouth, and leg regions of adult flies.(5)All nanopesticides at the highest doses—except for copper-based ones—had significant pro-oxidant effects in *Drosophila*.(6)High doses of pesticides caused significant changes in glutathione and lipid peroxidation formation in *Drosophila* larvae.(7)Single-stranded and oxidative DNA damage was mediated by oxidative damage to the pyrimidine bases.(8)Nanopesticides and microparticle forms caused no mutagenic and/or recombinogenic effects.(9)All pesticides at the highest doses—except for copper-based ones—caused phenotypic variations across three generations of fruit flies.(10)Nanopesticides and microparticle forms significantly impaired the climbing and walking ability of adult flies.(11)Ingested nanopesticides found their way through the intestinal barrier into the intestinal lumen.(12)Pesticides/nanopesticides caused a significant increase in the expression of stress genes (*Hsp70* and *Hsp83*), antioxidant defense genes (*CAT* and *SOD2*), and a genomic integrity gene (*p53*).(13)Nanopesticides and microparticle pesticides induced significant impairment in the expression of the *DUOX*, *Hml*, *Muc68D*, and/or *PPO2* genes, as well as in the mRNA expression of *Ogg1*—a gene associated with DNA repair.

## Figures and Tables

**Figure 1 ijms-23-09121-f001:**
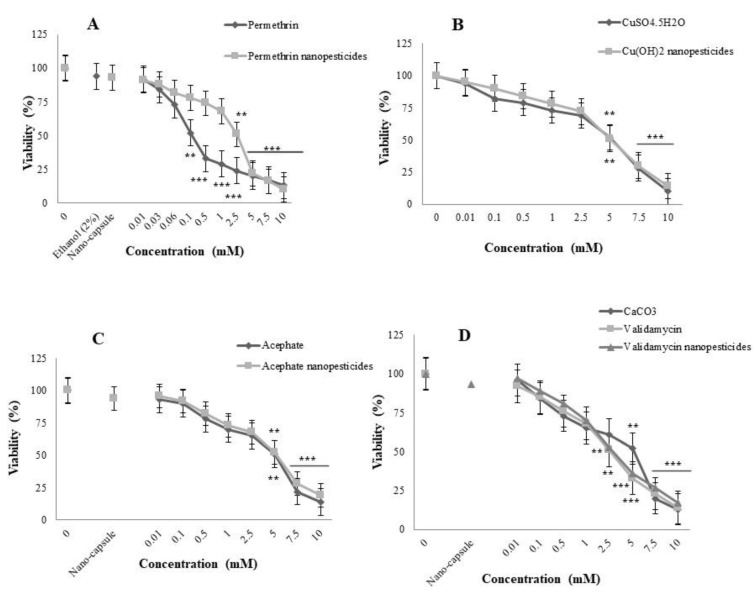
Toxicity of permethrin and permethrin nanopesticides (**A**), CuSO_4_·5H_2_O and Cu(OH)_2_ nanopesticides (**B**), acephate and acephate nanopesticides (**C**), and validamycin and validamycin nanopesticides (**D**) in *D. melanogaster*. Toxic effects were measured as loss of viability (egg-to-adult survival) relative to the control values. N = 5 vials per concentration, with 50 eggs per vial. The statistical approach was analyzed using the Mann–Whitney *U*-test; ******
*p* ≤ 0.01, *******
*p* ≤ 0.001 compared to controls.

**Figure 2 ijms-23-09121-f002:**
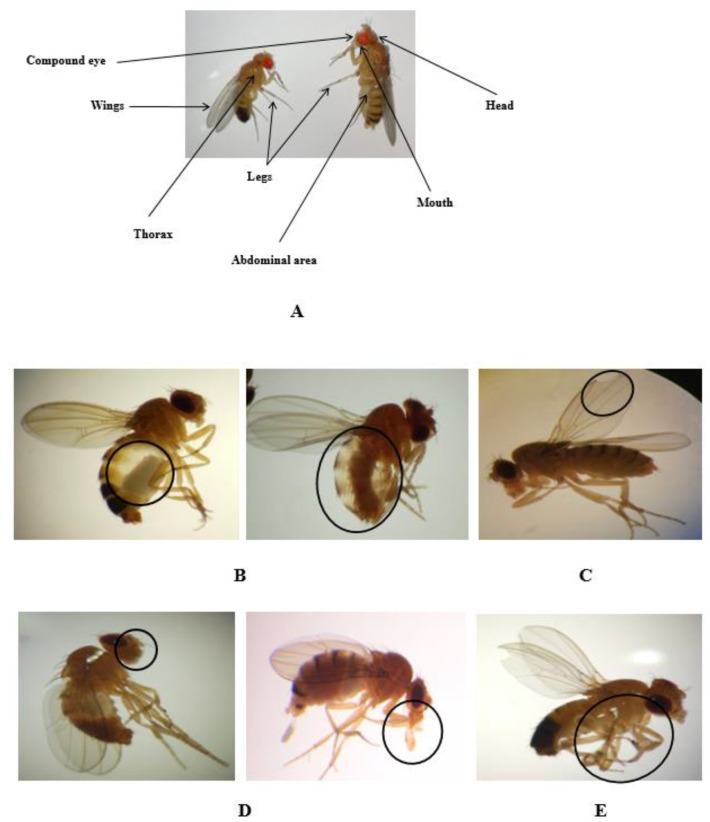
Adult morphology of *Drosophila* (**A**). Morphological alterations in the abdomen (**B**), wings (**C**), mouth (**D**), and legs (**E**) of adult *D. melanogaster* observed in permethrin (0.1 mM), permethrin nanopesticides (2.5 mM), acephate and acephate nanopesticides (5 mM), and validamycin and validamycin nanopesticides (2.5 mM). Black circles indicate the defective area in the image.

**Figure 3 ijms-23-09121-f003:**
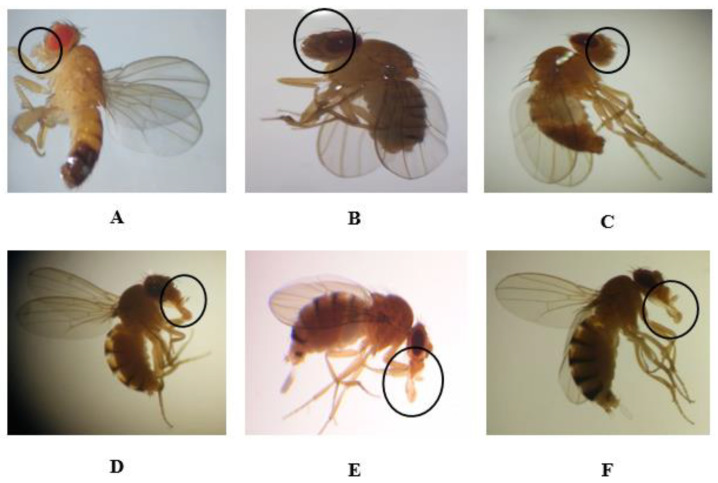
Normal (**A**) and abnormal mouth phenotypes (**B**–**F**) in the F0, F1, F2, and F3 generations obtained after treatment with permethrin (0.1 mM), permethrin nanopesticides (2.5 mM), acephate and acephate nanopesticides (5 mM), and validamycin and validamycin nanopesticides (2.5 mM) for *Drosophila* larvae. Black circles indicate the defective area in the image.

**Figure 4 ijms-23-09121-f004:**
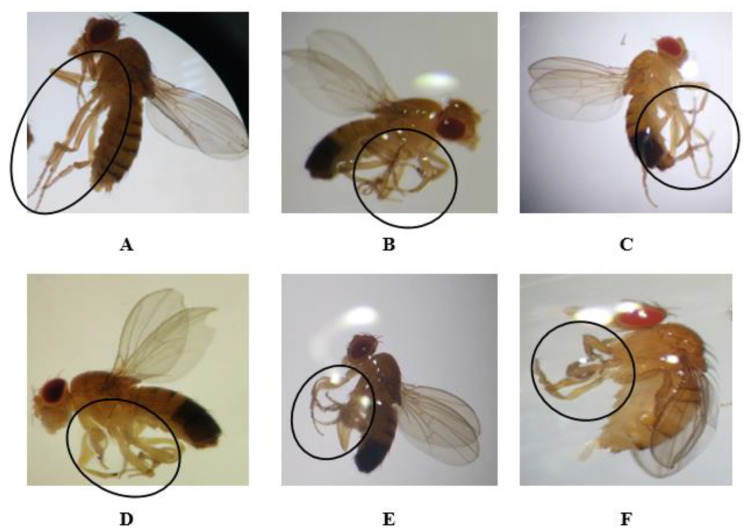
Normal (**A**) and abnormal leg phenotypes (**B**–**F**) in the F0, F1, F2, and F3 generations obtained after treatment with permethrin (0.1 mM), permethrin nanopesticides (2.5 mM), acephate and acephate nanopesticides (5 mM), and validamycin and validamycin nanopesticides (2.5 mM) for *Drosophila* larvae. Black circles indicate the defective area in the image.

**Figure 5 ijms-23-09121-f005:**
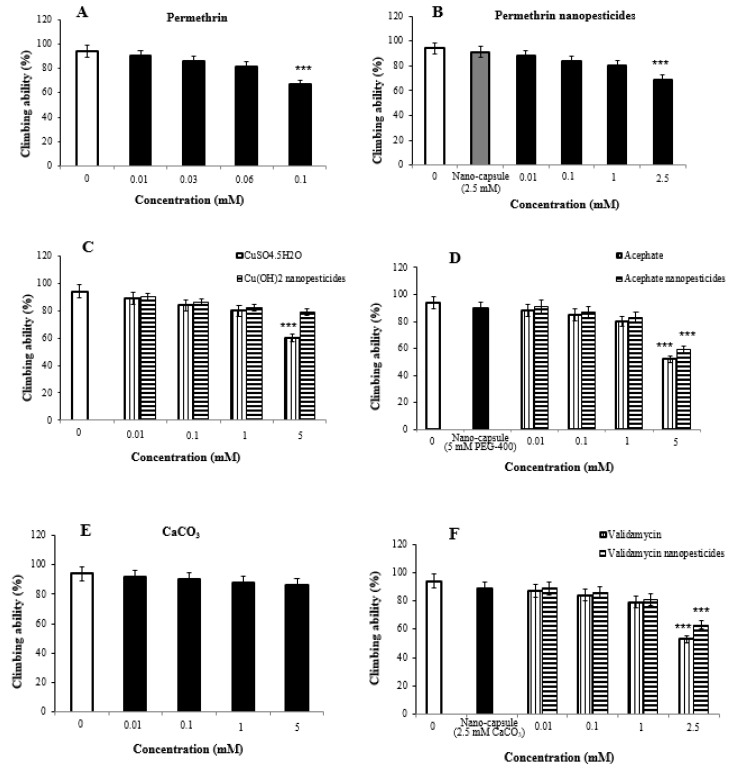
Climbing behavior of flies monitored after seven days of exposure to permethrin (**A**), permethrin nanopesticides (**B**), CuSO_4_·5H_2_O and Cu(OH)_2_ nanopesticides (**C**), acephate and acephate nanopesticides (**D**), CaCO_3_ (**E**), and validamycin and validamycin nanopesticides (**F**), recorded after 10 s. Data represent the mean ± standard error (SE) of the mean; *******
*p* ≤ 0.001 when compared to the negative control by one-way ANOVA.

**Figure 6 ijms-23-09121-f006:**
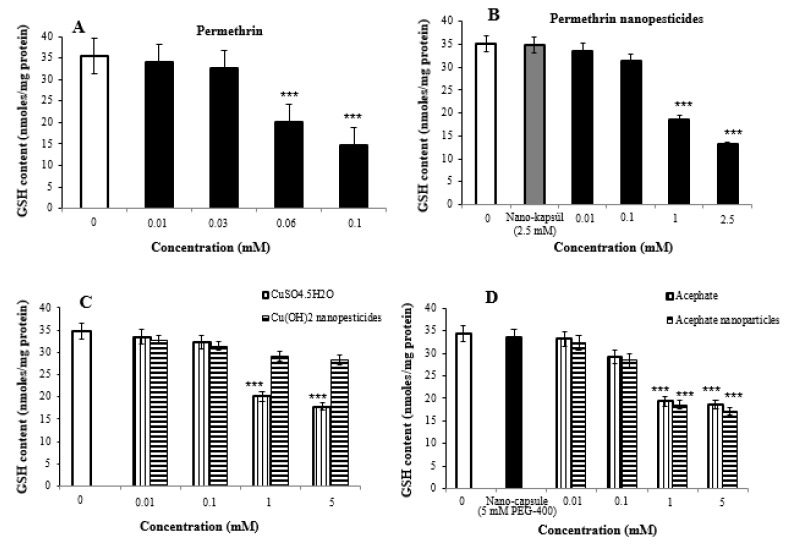
GSH contents in untreated controls (distilled water) and in permethrin (**A**), permethrin nanopesticides (**B**), CuSO_4_·5H_2_O and Cu(OH)_2_ nanopesticides (**C**), acephate and acephate nanopesticides (**D**), CaCO_3_ (**E**), and validamycin and validamycin nanopesticides (**F**) given to third-instar larvae of *Drosophila*. Each point represents a mean of 5 replicates ± SE; *******
*p* ≤ 0.001 when compared to the negative control using Student’s *t*-test.

**Figure 7 ijms-23-09121-f007:**
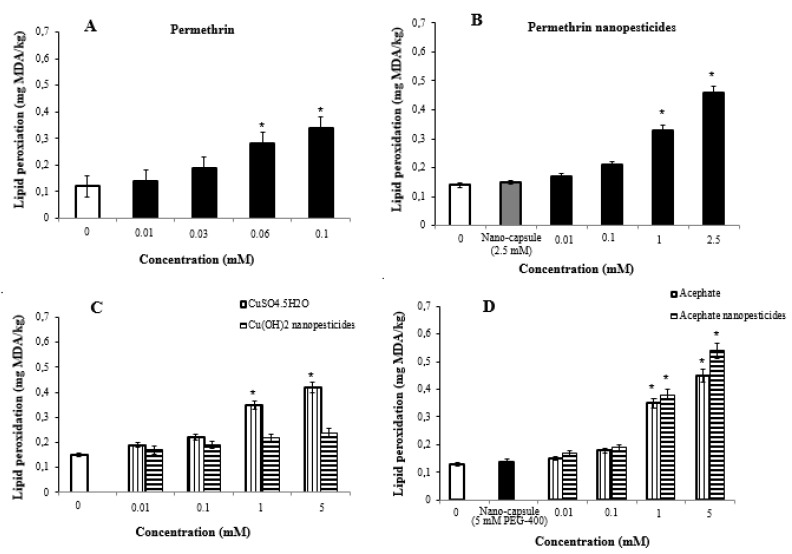
Lipid peroxidation, measured by TBARS (thiobarbituric-acid-reactive substances) accumulation in *Drosophila* treated with different concentrations of permethrin (**A**), permethrin nanopesticides (**B**), CuSO_4_·5H_2_O and Cu(OH)_2_ nanopesticides (**C**), acephate and acephate nanopesticides (**D**), CaCO_3_ (**E**), and validamycin and validamycin nanopesticides (**F**). Each point represents a mean of 3 replicates ± SE; *****
*p* ≤ 0.05 when compared to the negative control using one-way ANOVA.

**Figure 8 ijms-23-09121-f008:**
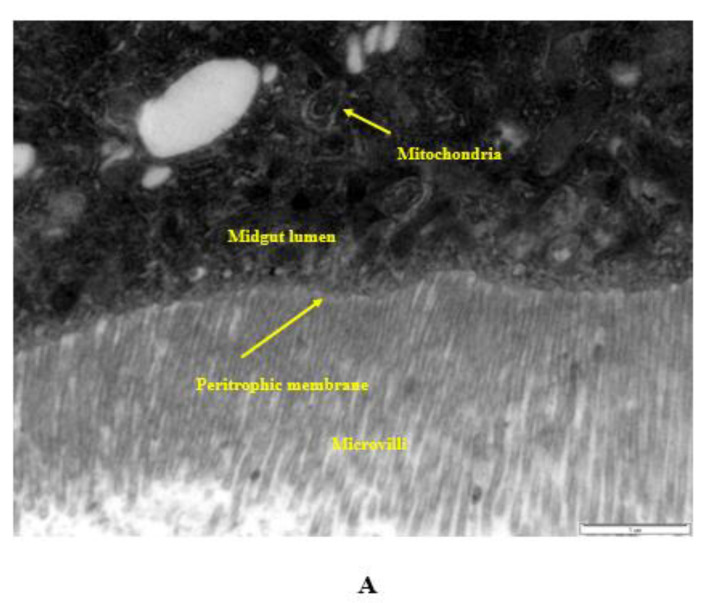
Internalization via the intestinal barrier: Intestinal lumen of *Drosophila* larvae (**A**). Permethrin nanopesticides (**B**), Cu(OH)_2_ nanopesticides (**C**), acephate nanopesticides (**D**), and validamycin nanopesticides (**E**) in the intestinal lumina of *Drosophila* larvae after oral ingestion. Permethrin nanopesticides perforated the first line of defense of the intestinal barrier—the peritrophic membrane—during their entry into the midgut cells.

**Figure 9 ijms-23-09121-f009:**
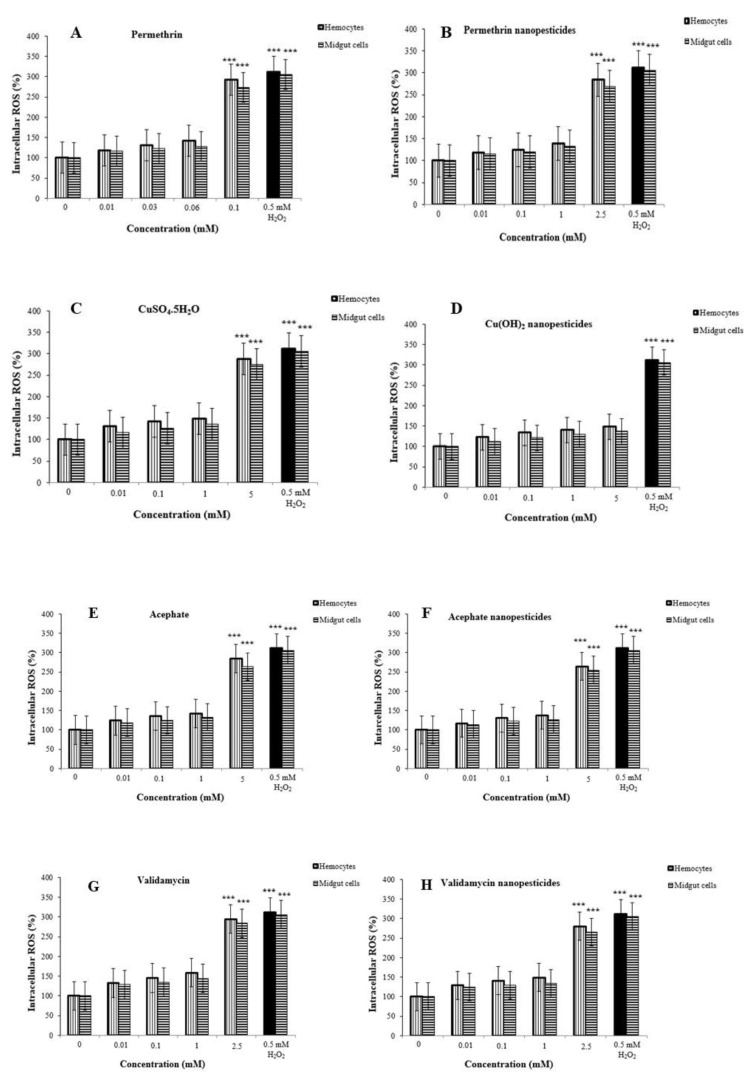
ROS production in hemocytes and midgut cells of third-instar untreated (0) and treated larvae exposed to different concentrations of permethrin (**A**), permethrin nanopesticides (**B**), CuSO_4_·5H_2_O (**C**), Cu(OH)_2_ nanopesticides (**D**), acephate (**E**), acephate nanopesticides (**F**), validamycin (**G**), validamycin nanopesticides (**H**), and CaCO_3_ (**I**). Hemocytes and midgut cells were incubated with 5 μM DCFH-DA at 24 °C for 30 min and observed via fluorescence microscopy. The fluorescence intensity of the hemocytes and midgut cells of larvae treated with permethrin were quantified by ImageJ analysis; 0.5 mM H_2_O_2_ was used as a positive control; *******
*p* ≤ 0.001 when compared to the negative control using the Mann–Whitney *U*-test.

**Figure 10 ijms-23-09121-f010:**
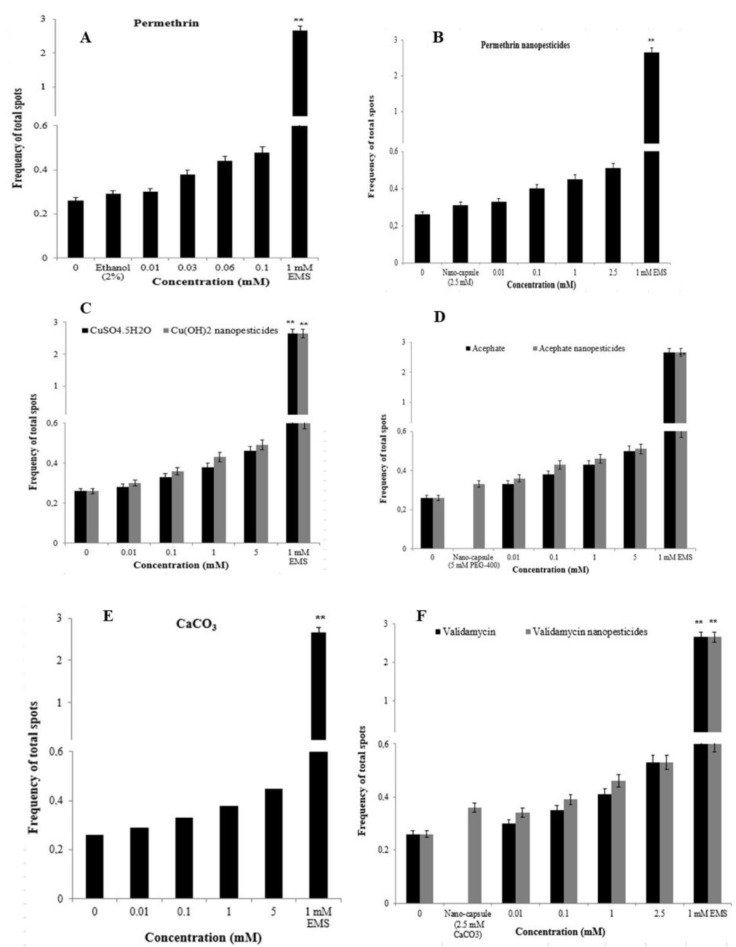
Induction of mutant clones in the wing spot assay: The frequency of total mutant spots per wing induced by permethrin (**A**), permethrin nanopesticides (**B**), CuSO_4_·5H_2_O and Cu(OH)_2_ nanopesticides (**C**), acephate and acephate nanopesticides (**D**), CaCO_3_ (**E**), and validamycin and validamycin nanopesticides (**F**) did not differ from the control frequency. As observed, EMS (1 mM) induced a clear increase in the frequency of mutant clones. Data represent the mean ± standard error of the mean; ** *p <* 0.01 compared with untreated larvae (Mann–Whitney *U*-test).

**Figure 11 ijms-23-09121-f011:**
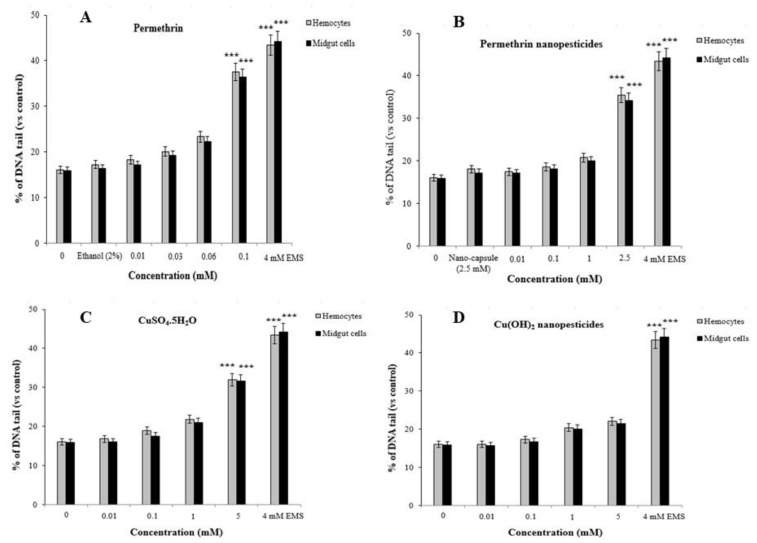
Genotoxic effects of permethrin (**A**), permethrin nanopesticides (**B**), CuSO_4_·5H_2_O (**C**), Cu(OH)_2_ nanopesticides (**D**), acephate (**E**), acephate nanopesticides (**F**), validamycin (**G**), validamycin nanopesticides *(***H***)*, and CaCO_3_ (**I**) in the comet assay. Results indicate the % DNA tail induced after the larvae were exposed to different doses of permethrin and permethrin nanopesticides for 24 h (three replicates were carried out, and 100 randomly selected cells were analyzed per treatment). Net oxidative damage induction in hemocytes and midgut cells after exposure to permethrin and permethrin nanopesticides at doses of 0.1 and 2.5 mM during the larval stage; effects induced by buffer exposure were subtracted from those obtained after enzyme treatments (Endo III and Fpg) (**J**–**M**). Data represent the mean ± standard error (SE) of the mean. EMS (4 mM) was used as a positive control; *****
*p* ≤ 0.05, *******
*p* ≤ 0.001 when compared to the negative control using Student’s *t*-test.

**Figure 12 ijms-23-09121-f012:**
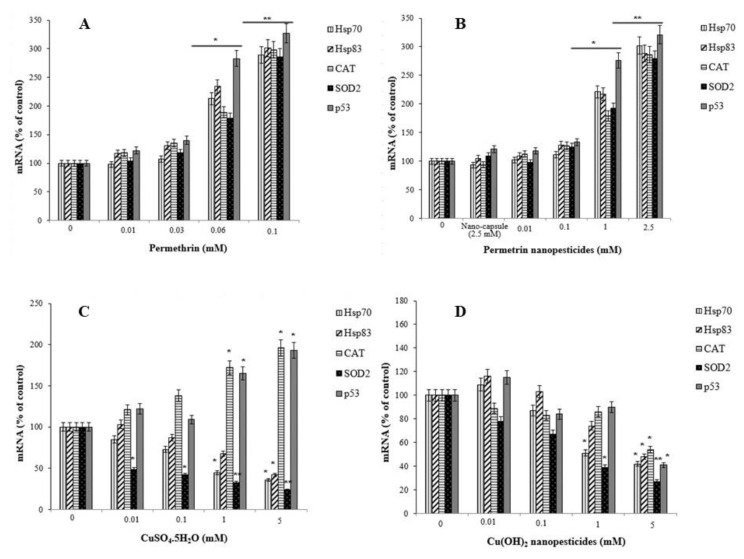
The expression of *Hsp70*, *Hsp83*, *CAT*, *SOD2*, and *p53* in 96 h larvae induced after exposure to different doses of permethrin (**A**), permethrin nanopesticides (**B**), CuSO_4_·5H_2_O (**C**), Cu(OH)_2_ nanopesticides (**D**), acephate (**E**), acephate nanopesticides (**F**), validamycin (**G**), validamycin nanopesticides (**H**), and CaCO_3_ (**I**). The expressions were normalized using β-actin, and are presented compared to control values. Data represent the mean ± standard error of the mean of six independent experiments; * *p* < 0.05, ** *p* < 0.01 (Student’s *t*-test).

**Figure 13 ijms-23-09121-f013:**
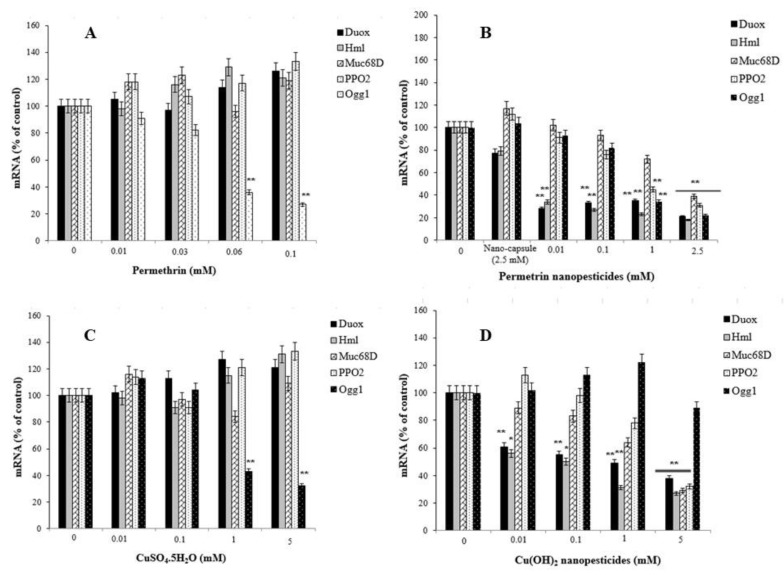
The expression of genes related to midgut injury and hemocyte interactions (Duox, Hml, Muc68D, and PPO2) and DNA repair (Ogg1) in *D. melanogaster* larvae exposed to permethrin (**A**), permethrin nanopesticides (**B**), CuSO_4_·5H_2_O (**C**), Cu(OH)_2_ nanopesticides (**D**), acephate (**E**), acephate nanopesticides (**F**), validamycin(**G**), validamycin nanopesticides (**H**), and CaCO_3_ (**I**). The expressions were normalized using β-actin. Data represent the mean ± standard error of the mean of six independent experiments; * *p* < 0.05, ** *p* < 0.01 (Student’s *t*-test).

## Data Availability

Not applicable.

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
