# Peer review of "Hazard Assessment of the Effects of Acute and Chronic Exposure to Permethrin, Copper Hydroxide, Acephate, and Validamycin Nanopesticides on the Physiology of Drosophila: Novel Insights into the Cellular Internalization and Biological Effects"

_ijms, 2022, doi:10.3390/ijms23169121_

Round 1
Reviewer 1 Report
The manuscript entitled “Hazard Assessment of Acute and Chronic Exposure of Permethrin, Copper hydroxide, Acephate and Validamycin Nanopesticides on the Physiology of Drosophila: Novel Insights into the Cellular Internalization and Biological Effects” is an important study use of pesticides to control pest (insect) resistance further contaminates terrestrial and aquatic ecosystems, along with toxic residues in foods grown for human or animal consumption. The annual death toll due to ingestion of pesticides through such foods is around 20,000, since they interact with the microbiome in the human gastrointestinal tract, causing digestive problems, lung cancer, and hormonal imbalances.
The introduction is relevant but must include new references. The discussion, in the light of results and knowledge, is relevant.
Your manuscript will not be accepted unless both the technical and grammatical revisions have been made successfully.
Based on these comments, I recommend a moderate revision of analytical aspects of this manuscript before final decision about its acceptance.
Moderate comment:
Introduction:
Rewrite this part with new references it is a very old research (e.g., Anjali, C.H.; et al., 2010 and Paul, A.; et al., 2006)
See reference: Recent study (Parrino et al., 2019 - 2021) showed the role of produce biomass protein rich with modern techniques.
It's necessary to insert new references about the use of natural component, see below reference of trace elements:
Vincenzo Parrino, Gregorio Costa, Alessia Giannetto, Giuseppe De Marco, Gaetano Cammilleri, Ümit Acar, Giuseppe Piccione, Francesco Fazio - Trace elements (Al, Cd, Cr, Cu, Fe, Mn, Ni, Pb and Zn) in Mytilus galloprovincialis and Tapes decussatus from Faro and Ganzirri Lakes (Sicily, Italy): Flow cytometry applied for hemocytes analysis. JOURNAL OF TRACE ELEMENTS IN MEDICINE AND BIOLOGY, 68, 126870; https://doi.org/10.1016/j.jtemb.2021.126870; (2021).
Parrino Vincenzo, Costa Gregorio, Cannavà Carmela, Fazio Enza, Bonsignore Martina, Saoca Concetta, Piccione Giuseppe, Fazio Francesco - Flow cytometry and micro-Raman spectroscopy: Identification of hemocyte populations in the mussel Mytilus galloprovincialis (Bivalvia: Mytilidae) from Faro Lake and Tyrrhenian Sea (Sicily, Italy). - FISH AND SHELLFISH IMMUNOLOGY, vol. 87, p. 1-8, ISSN: 1050-4648; - doi: 10.1016/j.fsi.2018.12.067; (2019).
2. Methods
2.1 Chemicals
I do not understand why the authors used in the chemicals utilized during the experiments were acquired from Sigma Chemical Co. For negative controls, we used sterile distilled water, ethanol 244 (2%) as the medium in which permethrin was dissolved; nanocapsules (2.5 mM) for the preparation of permethrin nanopesticide; nanocapsules (5 mM PEG-400) for the preparation of acephate nanopesticide; and nanocapsules (2.5 mM CaCO3) for the preparation of validamycin nanopesticide. This paragraph has been changed and more information has been included in the text…please rewrite better, on basis on the results obtained and the use of the protocol of the experiment developed by you with due care?
Which photoperiod was used for the experiments?
Statistical:
The statistical analysis used in all samples is appropriate.
3. Results:
The authors should be reduce this part, please the results shown in figures.
4. Discussion
This part is very long, reduce it and discuss only the obtained results.
However, abiotic conditions and exposure to pesticides, nanopesticides or various xenobiotic compounds can also activate different metabolic or physiological pathways depending on their way of acting. Therefore, the expression of several gene markers associated with cellular uptake, biological effects and changes can be influenced after exposure to pesticides / nanopesticides.
Author Response
RESPONSE TO THE REVIEWERS
Referee(s)' Comments to Author:
Reviewer: 1
Comments to the Author
The manuscript entitled “Hazard Assessment of Acute and Chronic Exposure of Permethrin, Copper hydroxide, Acephate and Validamycin Nanopesticides on the Physiology of Drosophila: Novel Insights into the Cellular Internalization and Biological Effects” is an important study use of pesticides to control pest (insect) resistance further contaminates terrestrial and aquatic ecosystems, along with toxic residues in foods grown for human or animal consumption. The annual death toll due to ingestion of pesticides through such foods is around 20,000, since they interact with the microbiome in the human gastrointestinal tract, causing digestive problems, lung cancer, and hormonal imbalances.
The introduction is relevant but must include new references. The discussion, in the light of results and knowledge, is relevant.
RESPONSE
Thank you very much for your valuable contribution. We thank for this reflection. We have added the following references in line with your comments.
[28] Parrino V.; Costa G.; Cannavà C.; Fazio E.; Bonsignore M.; Concetta S.; Piccione G.; Fazio F. Flow cytometry and micro-Raman spectroscopy: Identification of hemocyte populations in the mussel Mytilus galloprovincialis (Bivalvia: Mytilidae) from Faro Lake and Tyrrhenian Sea (Sicily, Italy). Fish Shellfish Immunol. 2019, 87, 1-8.
[29] Parrino V.; Costa G.; Giannetto A.; De Marco G.; Cammilleri G.; Acar Ü.; Piccione G.; Fazio F. Trace elements (Al, Cd, Cr, Cu, Fe, Mn, Ni, Pb and Zn) in Mytilus galloprovincialis and Tapes decussatus from Faro and Ganzirri Lakes (Sicily, Italy): Flow cytometry applied for hemocytes analysis. J. Trace Elem. Med. Biol. 2021, 68, 126870.
Your manuscript will not be accepted unless both the technical and grammatical revisions have been made successfully.
Based on these comments, I recommend a moderate revision of analytical aspects of this manuscript before final decision about its acceptance.
RESPONSE
Thank you very much for your valuable contribution and different point of view. We have revised the manuscript in line with your comments.
Moderate comment:
Introduction:
Rewrite this part with new references it is a very old research (e.g., Anjali, C.H.; et al., 2010 and Paul, A.; et al., 2006)
See reference: Recent study (Parrino et al., 2019 - 2021) showed the role of produce biomass protein rich with modern techniques.
It's necessary to insert new references about the use of natural component, see below reference of trace elements:
Vincenzo Parrino, Gregorio Costa, Alessia Giannetto, Giuseppe De Marco, Gaetano Cammilleri, Ümit Acar, Giuseppe Piccione, Francesco Fazio - Trace elements (Al, Cd, Cr, Cu, Fe, Mn, Ni, Pb and Zn) in Mytilus galloprovincialis and Tapes decussatus from Faro and Ganzirri Lakes (Sicily, Italy): Flow cytometry applied for hemocytes analysis. JOURNAL OF TRACE ELEMENTS IN MEDICINE AND BIOLOGY, 68, 126870; https://doi.org/10.1016/j.jtemb.2021.126870; (2021).
Parrino Vincenzo, Costa Gregorio, Cannavà Carmela, Fazio Enza, Bonsignore Martina, Saoca Concetta, Piccione Giuseppe, Fazio Francesco - Flow cytometry and micro-Raman spectroscopy: Identification of hemocyte populations in the mussel Mytilus galloprovincialis (Bivalvia: Mytilidae) from Faro Lake and Tyrrhenian Sea (Sicily, Italy). - FISH AND SHELLFISH IMMUNOLOGY, vol. 87, p. 1-8, ISSN: 1050-4648; - doi: 10.1016/j.fsi.2018.12.067; (2019).
RESPONSE
Thanks for the clarification. We have introduced a new paragraph in indicated section to emphasize this point. This read as:
“In some of the recent studies in the literature, hemocytes were used to monitor living organisms in nature for environmental conditions with new techniques (such as flow cytometry) [28,29].”
[28] Parrino V.; Costa G.; Cannavà C.; Fazio E.; Bonsignore M.; Concetta S.; Piccione G.; Fazio F. Flow cytometry and micro-Raman spectroscopy: Identification of hemocyte populations in the mussel Mytilus galloprovincialis (Bivalvia: Mytilidae) from Faro Lake and Tyrrhenian Sea (Sicily, Italy). Fish Shellfish Immunol. 2019, 87, 1-8.
[29] Parrino V.; Costa G.; Giannetto A.; De Marco G.; Cammilleri G.; Acar Ü.; Piccione G.; Fazio F. Trace elements (Al, Cd, Cr, Cu, Fe, Mn, Ni, Pb and Zn) in Mytilus galloprovincialis and Tapes decussatus from Faro and Ganzirri Lakes (Sicily, Italy): Flow cytometry applied for hemocytes analysis. J. Trace Elem. Med. Biol. 2021, 68, 126870.
- Methods
2.1 Chemicals
I do not understand why the authors used in the chemicals utilized during the experiments were acquired from Sigma Chemical Co. For negative controls, we used sterile distilled water, ethanol 244 (2%) as the medium in which permethrin was dissolved; nanocapsules (2.5 mM) for the preparation of permethrin nanopesticide; nanocapsules (5 mM PEG-400) for the preparation of acephate nanopesticide; and nanocapsules (2.5 mM CaCO3) for the preparation of validamycin nanopesticide. This paragraph has been changed and more information has been included in the text…please rewrite better, on basis on the results obtained and the use of the protocol of the experiment developed by you with due care?
RESPONSE
We thank for this reflection. We have revised 2.1 Chemicals section in line with your comments. This read as:
“Ethyl methanesulfonate (EMS, CAS No. 62-50-0), hydrogen peroxide (H2O2, CAS No. 7722-84-1), acephate (C4H10NO3PS, ≥ 98), permethrin (C21H20Cl2O3, ≥ 90), hexadecyl trimethyl ammonium bromide (C19H42BrN, ≥98), phosphatidylcholine (from soybean lecithin) (C42H80NO8P, 40%) and ammonium glycyrrhizate (C42H62O16.NH3, ≥70) were purchased from Sigma Chemical Co. (St. Louis, MO, USA).”.
“For negative control groups, we used sterile distilled water, ethanol (2%), nanocapsules (2.5 mM), nanocapsules (5 mM PEG-400), and nanocapsules (2.5 mM CaCO3) for the preparation of CuSO4.5H2O and Cu(OH)2 nanopesticide, permethrin, permethrin nanopesticide, acephate nanopesticide, and validamycin nanopesticide, respectively.”
Which photoperiod was used for the experiments?
RESPONSE
We thank for this reflection. We added the relevant information as in a 12:12 hours light:dark cycle.
Statistical:
The statistical analysis used in all samples is appropriate.
RESPONSE
We thank the Reviewer for the positive feedback. Thank you very much for your valuable contribution and different point of view.
- Results:
The authors should be reduce this part, please the results shown in figures.
RESPONSE
Thank you very much for your valuable contribution and different point of view.
- Discussion
This part is very long, reduce it and discuss only the obtained results.
However, abiotic conditions and exposure to pesticides, nanopesticides or various xenobiotic compounds can also activate different metabolic or physiological pathways depending on their way of acting. Therefore, the expression of several gene markers associated with cellular uptake, biological effects and changes can be influenced after exposure to pesticides / nanopesticides.
RESPONSE
We appreciate the response. Thank you very much for your valuable contribution.

Reviewer 2 Report
My comments are as follows:
- This paper is sound and thorough but dense - as I was reading, it seemed to me that the synthesis and testing of the compounds could be its own paper and the effect on flies its own as well.
- The title could be shortened so as to help with searchability; this is where I first though this could be two papers. (Or perhaphs Synthesis and assessment of four nanopesticides on viability and physiology of Drosophila? But there are effects you describe that I think the paper should lead with!)
- Line 41 "desperately" doesn't come across as scientific
- The introduction is lengthy so my advice is to include a small figure up front with the flow chart for synthesis and testing of the compounds with the flies that the paper goes through.
- What larval stage is used for the oxidative stress assay? I may have missed it but as a dev bio it might matter.
- For the climbing assay, did the group anesthetize with CO2 beforehand? This requires a longer acclimation period.
- Lipid peroxidation and intestinal crossing mention 3rd instar but I didn't see it prior to these.
- The methods are wonderfully detailed but I wonder if some of these details belong in SI and the figures down there belong in the main text?
- Figure 1, 5, 6, 7, 9, 10, 12, 13 labels of axes and bar differences are a little hard to read. Might be the program used, but it takes away from your interesting results, especially in figures 9 and 10.
- Figure 8 contains great information but the boxes are not formatted and the text is hard to read.
- Figure 11 has the same issue with the text and bars and is not formatted similar to Figure 8 - can this one be split into two, perhaps one is the hemocyte assay and the next shows the ox damage?
- For me, the most interesting part is the gene expression changes you noted in some important and versatile genes! This and the DNA repair need to be highlighted - can you lead with these things instead of titling the paper 'exposure'?
- At the segue between results and conclusions, you should make a figure that recaps all of the findings overall. Even simple arrows and upside down flies will work. This paper is dense and you don't want any of these important results getting drowned out.
- Line 1137 'in' should be removed; also your genus and species should be italicized throughout; quick fix
- Your discussion contains quite a bit of material on toxicity in animals and would benefit from some kind of visual - this paper presents a ton of useful data but lacks a visual component to tie it together.
- The conclusion nicely lays out 12 findings from this paper. My advice is to lead with the most useful or interesting conclusions back in the title and to make a figure, even super simple, that shows body parts and absorption arrows and disrupted gene pathways with simple shapes.
- I mentioned up top my opinion that the SI figures should be in the main text and some of the most precise details about the discovery and history (introduction) and synthesis (methods) could move back to SI.
Reviewer 3 Report
The paper is very extensive, all methods are described in detail, the results are clearly presented and explained, and the discussion is systematically and meaningfully written. The conclusions are derived from the obtained research, and I believe that this paper meets all the requirements for publication.
Author Response
RESPONSE TO THE REVIEWERS
Referee(s)' Comments to Author:
Reviewer: 2
Comments to the Author
The paper is very extensive, all methods are described in detail, the results are clearly presented and explained, and the discussion is systematically and meaningfully written. The conclusions are derived from the obtained research, and I believe that this paper meets all the requirements for publication.
RESPONSE
We thank the Reviewer for the positive feedback. Thank you very much for your valuable contribution.
